# A comprehensive analysis of coregulator recruitment, androgen receptor function and gene expression in prostate cancer

Song Liu[1†], Sangeeta Kumari[2†], Qiang Hu[1], Dhirodatta Senapati[2], Varadha Balaji Venkadakrishnan[2], Dan Wang[1], Adam D DePriest[3], Simon E Schlanger[2], Salma Ben-Salem[2], Malyn May Valenzuela[2], Belinda Willard[4], Shaila Mudambi[5], Wendy M Swetzig[6], Gokul M Das[6], Mojgan Shourideh[3], Shahriah Koochekpour[3], Sara Moscovita Falzarano[7], Cristina Magi-Galluzzi[7], Neelu Yadav[6], Xiwei Chen[1], Changshi Lao[8], Jianmin Wang[1], Jean-Noel Billaud[9], Hannelore V Heemers[2,10,11]*

[1]Department of Biostatistics and Bioinformatics, Roswell Park Cancer Institute, Buffalo, United States; [2]Department of Cancer Biology, Cleveland Clinic, Cleveland, United States; [3]Department of Cancer Genetics, Roswell Park Cancer Institute, Buffalo, United States; [4]Department of Research Core Services, Cleveland Clinic, Cleveland, United States; [5]Department of Cell Stress Biology, Roswell Park Cancer Institute, Buffalo, United States; [6]Department of Pharmacology and Therapeutics, Roswell Park Cancer Institute, Buffalo, United States; [7]Department of Anatomic Pathology, Cleveland Clinic, Cleveland, United States; [8]Institute for Nanosurface Science and Engineering, Shenzhen University, Shenzhen, China; [9]QIAGEN Bioinformatics, Redwood City, United States; [10]Department of Urology, Cleveland Clinic, Cleveland, United States; [11]Department of Hematology/Medical Oncology, Cleveland Clinic, Cleveland, United States

*For correspondence: heemerh@ccf.org

[†]These authors contributed equally to this work

Competing interests: The authors declare that no competing interests exist.

**Abstract** Standard treatment for metastatic prostate cancer (CaP) prevents ligand-activation of androgen receptor (AR). Despite initial remission, CaP progresses while relying on AR. AR transcriptional output controls CaP behavior and is an alternative therapeutic target, but its molecular regulation is poorly understood. Here, we show that action of activated AR partitions into fractions that are controlled preferentially by different coregulators. In a 452-AR-target gene panel, each of 18 clinically relevant coregulators mediates androgen-responsiveness of 0–57% genes and acts as a coactivator or corepressor in a gene-specific manner. Selectivity in coregulator-dependent AR action is reflected in differential AR binding site composition and involvement with CaP biology and progression. Isolation of a novel transcriptional mechanism in which WDR77 unites the actions of AR and p53, the major genomic drivers of lethal CaP, to control cell cycle progression provides proof-of-principle for treatment via selective interference with AR action by exploiting AR dependence on coregulators.
DOI: https://doi.org/10.7554/eLife.28482.001

## Introduction

The androgen-activated androgen receptor (AR) is both the major driver of prostate cancer (CaP) progression and the main target for treatment of metastatic CaP. An initial remission after AR-targeting androgen deprivation therapy (ADT) almost inevitably results in recurrence because CaP cells acquire resistance to ADT and continue to rely on AR activity (*Karantanos et al., 2015*; *Dai et al.,*

**eLife digest** Prostate cancer is the second leading cause of cancer deaths in men in the Western world. Almost all of these deaths happen when the main treatment for advanced prostate cancers stops working. The treatment, known as androgen deprivation therapy, targets a protein called the androgen receptor. This receptor is activated when it binds to signaling molecules and, once active, it switches on genes that encourage the cancer cells to grow. Androgen deprivation therapy blocks the androgen receptor from interacting with the signaling molecules; however, this treatment eventually fails because the receptor finds other ways to remain active in prostate cancer.

Increasing the survival of patients with prostate cancer will depend on new treatments that can inhibit androgen receptors that no longer respond to androgen deprivation therapy. The androgen receptor's ability to switch on genes could be another target for prostate cancer therapy – though not enough was known about the way this ability is regulated and how it controls the progression of prostate cancer. Liu, Kumari et al. set out to better define how this ability drives the growth of prostate cancer.

The androgen receptor needs to interact with other proteins, known as coregulators, to work, and Liu, Kumari et al. developed an assay that examines, all at the same time, how important 18 such coregulators are for more than 400 genes that are regulated by the androgen receptor. This revealed that the coregulators did not all affect the same genes and that each coregulator tended to help activate sets of genes associated with a specific aspect of the biology of prostate cancer cells.

Liu, Kumari et al. also discovered previously unknown interactions between androgen receptors, coregulators and other proteins that were responsible for the specific associations between genes and corregulators. The most important of these new interactions was one between the androgen receptor, the coregulator WDR77, and a protein called p53. These interactions are enriched in prostate cancers, including those that do not respond to androgen deprivation therapy, where they promote cancer growth.

These findings lay the foundation to develop new drugs that interfere with the interactions between the androgen receptor and other proteins that are most important for the progression of advanced prostate cancers. Other researchers have already shown that it is possible to develop such drugs – though further testing is needed before any new treatments begin to help prostate cancer patients who no longer respond to androgen deprivation therapy.

DOI: https://doi.org/10.7554/eLife.28482.002

*2017*). With few exceptions, failure of ADT is responsible for the ~27,000 CaP deaths in the United States annually (*Siegel et al., 2017*).

Novel, alternative approaches to block the AR action that drives CaP to the lethal stage are highly sought. Current ADT prevents interaction between AR and its androgenic ligands, thus targeting the AR ligand-recognition function. AR's effector function as a transcription factor controls expression of Androgen Response Element (ARE)-driven genes (*Heinlein and Chang, 2004*; *Heemers and Tindall, 2007*) and ultimately dictates androgen-regulated CaP cell behavior. Targeting AR's transactivation function may prevent or overcome resistance associated with current ADT and lead to more CaP-specific inhibition of AR activity (*Heemers, 2014*).

Interfering with AR action at the post-receptor level (*Heemers, 2014*) requires an understanding of the molecular mechanisms by which AR controls expression of target genes that drive CaP progression. Increasingly sophisticated systems biology and bioinformatics techniques have provided insights to the AR-dependent transcriptome, the AR cistrome, and the composition of genomic AR binding sites (ARBSs) in CaP cells (*Horie-Inoue and Inoue, 2013*; *Mills, 2014*). Combined results from these endeavors suggest that gene specificity may exist in AR control over androgen-dependent gene expression. This possibility is in line with previous reports that coregulators, master regulators of transcription that are recruited to ARE-bound AR, preferentially control androgen regulation of subsets of AR target genes (*Marshall et al., 2003*; *Ianculescu et al., 2012*; *Xu et al., 2009*; *Heemers et al., 2009*).

Coregulators have long been of interest as therapeutic targets for CaP (*Heemers and Tindall, 2005*; *Chmelar et al., 2007*). Expression of 50 of the ~200 AR-associated coregulators is deregulated in clinical CaP specimens. Such aberrant expression often correlates with aggressive disease and poor outcome (*Heemers and Tindall, 2010*) and is one of the mechanisms that lead to resistance to conventional ADT. Many coregulators possess enzymatic functions for which inhibitor(s) are already available (e.g.(*Wang et al., 2011*)). Combinations of chemical library-screening and Chem-Seq approaches have identified novel coregulator-targeting drugs (*Jin et al., 2014*) while advances in peptidomimetics and multivalent peptoid conjugates are allowing for disruption of selective coregulator-AR interactions (*Ravindranathan et al., 2013*; *Wang et al., 2016*).

The true potential of coregulators as alternative targets to block AR action in CaP and their contribution to AR-dependent transcription that drives CaP progression, however, remains unknown. The few studies so far (*Marshall et al., 2003*; *Ianculescu et al., 2012*; *Xu et al., 2009*; *Heemers et al., 2009*) have taken into account only the impact of coregulators on androgen regulation of a handful of well-characterized exogenous or endogenous ARE-driven genes, or genome-wide gene expression profiles. In most cases investigations have been limited to the study of a single coregulator, without considering its relevance to clinical CaP progression, or redundancy or cooperativity between coregulators interacting with ARE-bound AR.

Here, we use an integrated approach to systematically define the contribution of 18 clinically relevant coregulators to androgen responsiveness of 452 *bona fide* AR target genes. Our results demonstrate a previously unrecognized level of gene specificity and context-dependence in reliance of AR target gene expression on coregulators, and the corresponding AR target gene sets contribute differentially to CaP initiation and progression. Analysis of the molecular basis and associated cell biology of coregulator-dependent AR target gene expression indicates transcriptional codes exist in which AR cooperates with select coregulator(s) and transcription factors to control transcription of a subset of its target genes. Our identification of a WDR77-dependent functional interaction between AR and p53 provides a rationale for a coregulator-dependent alternative to target for therapy the major drivers of lethal CaP progression.

## Results

### Isolation of a *bona fide* AR target gene signature

A systematic analysis of the role for coregulators in regulation of AR function requires a sizable set of ARE-driven AR target genes that can be interrogated coordinately. System biology approaches per se do not provide an unambiguous signature of AR target genes. Expression profiles of androgen-regulated genes do not distinguish between direct AR target genes, which are androgen-responsive because of direct AR-ARE interaction, and indirect AR target genes, which are androgen-regulated secondary to the action of a direct AR target gene. ChIP-chip, ChIP-Seq and ChIP-exo studies document androgen-dependent recruitment of AR throughout the genome but are prone to artifacts, only a small fraction of isolated ARBSs undergo independent ChIP validation, and the association between an ARBS(s) and androgen regulation of an adjacent gene often remains elusive.

We reasoned that integrating information on the genome-wide location of ARBSs, transcriptional start site (TSS) position, and androgen-responsive gene expression would result in identification of *bona fide* direct AR target genes. ARBSs present within 300 Kb of TSSs of RefSeq genes after androgen treatment (*Wang et al., 2009*) of AR-positive LNCaP cells were retrieved, and the overlap between the corresponding RefSeq gene list and androgen-dependent CaP gene expression profiles (*Wang et al., 2009*; *DePrimo et al., 2002*; *Nelson et al., 2002*; *Segawa et al., 2002*; *Febbo et al., 2005*; *Velasco et al., 2004*; *Ngan et al., 2009*; *Waghray et al., 2001*; *Xu et al., 2001*) was defined. This approach narrowed down a set of 12,629 ARBSs to 900 putative direct AR target genes (*Figure 1—figure supplement 1*). A custom (Agilent 8 × 15 k) gene expression oligoarray was developed to assess simultaneously expression of these genes (*Supplementary file 1*, panel A). Oligoarray performance was assessed using RNA from LNCaP cells treated with the synthetic androgen R1881 or vehicle. Prior to oligoarray assay, real-time RT-PCR analysis of AR target genes *PSA*, *FN1*, and *SCAP* (*Cleutjens et al., 1996*; *Cleutjens et al., 1997*; *Heemers et al., 2004*; *Bolton et al., 2007*) verified androgen-responsiveness of cells and RNA quality. Stimulation of LNCaP cell growth under these treatment conditions was verified via Ki67 immunocytochemistry and trypan blue

exclusion experiments (*Figure 1—figure supplement 2*). Oligoarray data revealed 452 genes with at least 2-fold change in expression in response to androgens (*Supplementary file 1*, panel B). These genes included well-characterized AR target genes such as *PSA* (*Cleutjens et al., 1996*; *Cleutjens et al., 1997*), *TMPRSS2* (*Wang et al., 2007*), *FN1* (*Bolton et al., 2007*) and *SERPINB5* (*Zhang et al., 1997*), as well as genes that are less readily recognized as AR target genes, e.g. *RALB, MPRIP, GNL1, GNB4, GUCY1A3, ARHGAP11A, WASF3* and *RAB27A* (*Figure 1—figure supplement 1*). Androgen treatment increased expression of 241 (55%) of these genes, while reducing expression of the remaining 211. Androgen dependence and directionality of androgen regulation of >90% of the genes were also present in an independent AR-positive cell line, VCaP (*Korenchuk et al., 2001*) (*Supplementary file 2*, panel A, data not shown). The kinetics of androgen response (*Supplementary file 2*, panel B) was consistent with behavior of direct AR target genes, that is, androgen-induction of genes was notable at earlier time points (4 hr) than androgen-suppression (8 hr). ChIP verified androgen-dependent recruitment of AR to predicted AREs within ARBSs (*Figure 1—figure supplement 1*). No preference for particular chromosomes was noted, but consistent with other reports (*Horie-Inoue and Inoue, 2013*), ARBSs were predominantly located in enhancer regions and intergenic regions (*Figure 1—figure supplement 1*). The vast majority (84.6%) of ARBSs in the 452-AR target genes overlapped with androgen-induced H3K4me2 ChIP-Seq peaks, an epigenetic marker for active AR-dependent transcription (*He et al., 2010*). Cistrome motif analyses (*Liu et al., 2011*) of the DNA sequences that correspond to ARBSs demonstrated significant enrichment for the consensus AR binding motif. The top 10 enriched motifs also included the highly similar and sometimes interchangeable binding motifs for related glucocorticoid and progesterone nuclear receptors, as well as AR-interacting pioneering factor FoxA1 and the related FoxA2, general transcription factor GTF2A1 and transcription factors (TFs) such as STAT1 that are known to interact with ARE-bound AR (*Figure 1—figure supplement 1*). Ingenuity Pathway Analysis (*Krämer et al., 2014*) (IPA) indicated significant enrichment for cell functions associated with cancer; cell growth (specifically of CaP cells), death and movement, lipid metabolism (*Supplementary file 3*, panel A). These processes have been independently reported to be under androgen control in CaP cells (*Dehm and Tindall, 2006*; *Heemers et al., 2006*). These diverse and complementary analyses of the 452 gene signature indicate that the isolated 452 gene set is suitable for the proposed studies.

## AR-associated coregulators with relevance to CaP progression

AR-associated coregulators that are most relevant to CaP aggressiveness will provide the most clinically useful insights. At the onset of this study, 181 coregulators that interact physically and functionally with AR had been identified (*Supplementary file 1*). For 51 AR-associated coregulators protein expression was deregulated between CaP and benign prostate. Differential expression of 22 of these 51 correlated with more aggressive CaP features and shorter disease-free survival after prostatectomy (*Heemers and Tindall, 2010*). These 22 coregulators, which likely represent critical contributors to AR activity in CaP progression, were analyzed further. LNCaP cells were transfected with specific siRNAs to individually silence each coregulator. Silencing efficacy and specificity were verified using real-time RT-PCR. No adverse effects on cell death and cell appearance were seen after knock-out of any of the 22 coregulators. Silencing of GAK, HIP1, RAD9A or SMAD3, however, did decrease markedly AR protein expression (*Supplementary file 4*). These 4 coregulators were excluded from subsequent experiments (*Figure 1A*) to avoid confounding interpretation of subsequent experiments.

## Gene-specific and context-dependent coregulator contribution

The relevance of the remaining 18 AR-associated coregulators (*Figure 1B*) to androgen responsiveness of AR target gene expression was determined next. Transfection of LNCaP cells with siRNAs that individually targeted each coregulator was combined with R1881 treatment (*Heemers et al., 2009*). For ≥95% of AR target genes (n > 35 tested), real-time RT-PCR analysis verified the oligoarray pattern of gene expression (*Supplementary file 2*). *Figure 1B* summarizes changes in androgen-responsiveness that occurred with silencing of each coregulator individually. The fraction of genes affected varied widely among different coregulators, ranging from 0% for RCHY1 siRNA transfection to 57% (258/452 genes affected) for p300 knock-down. These results demonstrate considerable gene-specific preference in the contribution of individual coregulators to the androgen regulation of

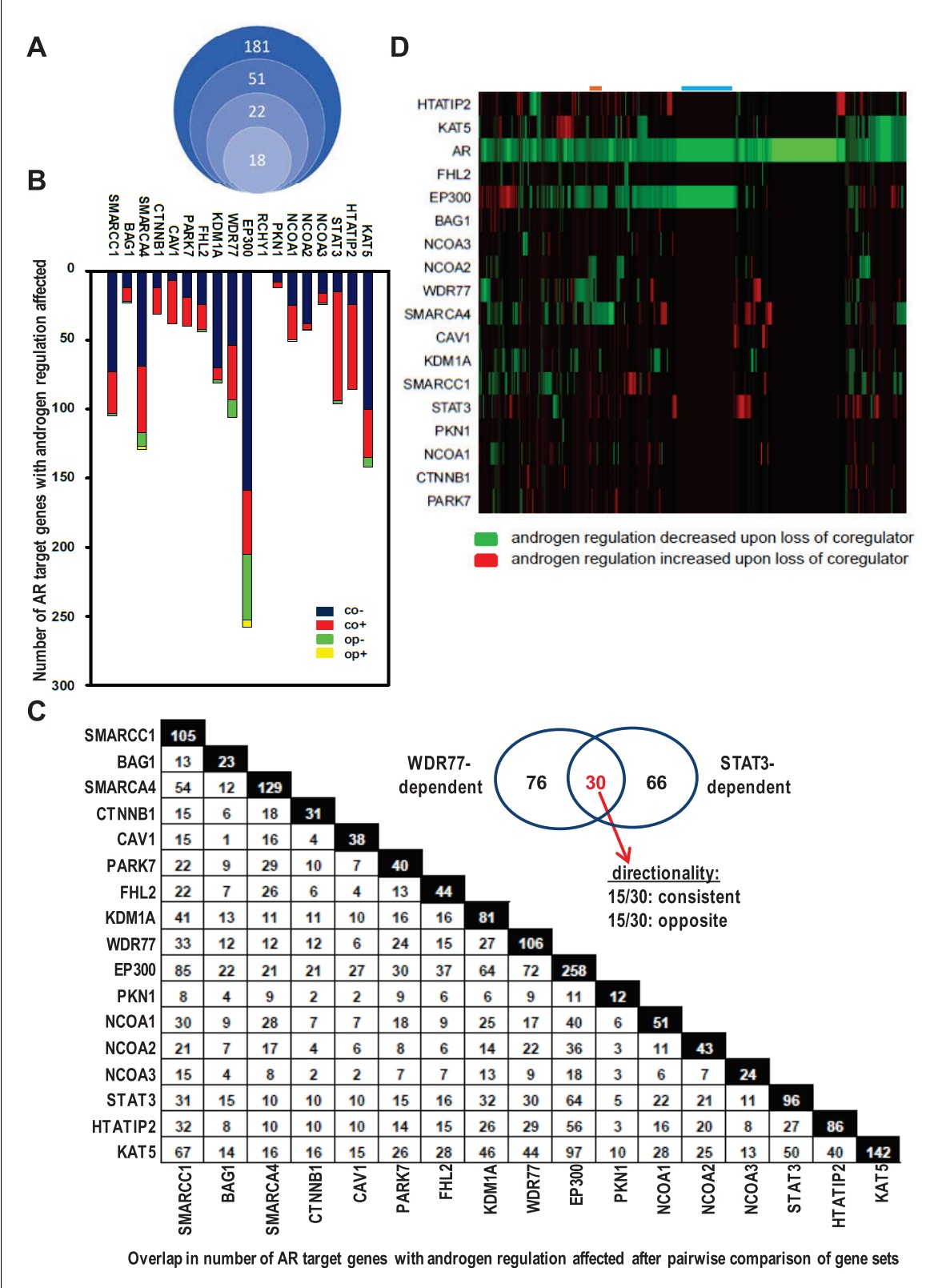

**Figure 1.** Contribution of 18 clinically relevant coregulators to androgen regulation of AR target gene expression. (**A**) 181 coregulators that interact physically and functionally with AR were considered for inclusion. 51 of the 181 coregulators demonstrate deregulated protein expression between CaP and benign prostate and these coregulators were withheld for further study. Differential expression of 22 of these 51 coregulators correlated with more aggressive CaP behavior. For these 22 coregulators, which likely represent critical contributors to AR activity in CaP progression, a siRNA screen was

*Figure 1 continued on next page*

*Figure 1 continued*

done in LNCaP cells to evaluate effects on AR expression, CaP cell morphology and CaP cell death. Silencing of 4 of the 22 coregulators decreased markedly AR protein expression. The latter 4 coregulators were excluded from subsequent experiments and the remaining 18 coregulators were withheld for subsequent studies on AR target gene expression. (B) Gene specificity and context-dependency of coregulator contribution to androgen regulation of AR target gene expression. co+, androgen regulation is increased (+) after loss of coregulator and direction of regulation remains consistent (co); co-, androgen regulation is decreased (-) after loss of coregulator and direction of regulation remains consistent (co); op+, androgen regulation is increased (+) after loss of coregulator but direction of regulation is opposite (op); op-, androgen regulation is decreased (-) after loss of coregulator and direction of regulation is opposite (op). Y-axis and bar lengths indicate the number of AR target genes for which androgen regulation is altered after silencing of each coregulator that is listed on the X-axis. Results reflect the effect of 48 hr treatment of LNCaP cells with 5nM R1881. R1881 or vehicle treatment was administered 42 hr after siRNA transfection. *Figure 1—source data 1* provides numerical information for data shown here. (C) Pairwise comparison between all coregulator-dependent AR target gene sets to determine overlap in number of genes. The X- and Y-axes mark the AR target gene sets for which androgen regulation is altered after knock-down of the individual coregulators that are listed on the axes. Black square indicates number of genes in the corresponding coregulator-dependent gene set. White square indicates the number of genes that overlap between 2 gene sets that intersect (left panel). *Figure 1—source data 2* provide the p-values for the significance of the overlap for the 136 pairwise comparisons. The Venn diagram shows the number of genes that are unique to or are shared by the WDR77- and STAT3-dependent AR target gene sets. Of the 30 genes that overlap between the 2 AR gene sets, 15 show consistent directionality of androgen regulation in both gene sets. The other 15 genes display opposite directionality of androgen regulation (right panel). (D) Unsupervised clustering of AR target genes (columns) based on coregulator dependence of affected AR target gene sets (rows). Green, gene for which androgen regulation decreases upon loss of coregulator (-); red, gene for which androgen regulation increases upon loss of coregulator (+); blue line, mutual exclusivity in coregulator dependence (EP300); orange line, cooperativity in coregulator dependence (EP300 and SMARCA4). AR silencing served as the control condition.

DOI: https://doi.org/10.7554/eLife.28482.003

The following source data and figure supplements are available for figure 1:

**Source data 1.** Gene specificity and context-dependency of coregulator contribution to androgen regulation of AR target gene expression.
DOI: https://doi.org/10.7554/eLife.28482.006
**Source data 2.** Summary of p-values for data presented in *Figure 1C*.
DOI: https://doi.org/10.7554/eLife.28482.007
**Figure supplement 1.** Isolation and validation of a 452 AR target gene signature.
DOI: https://doi.org/10.7554/eLife.28482.004
**Figure supplement 2.** The effect of 5nM R1881 on proliferation of LNCaP cells.
DOI: https://doi.org/10.7554/eLife.28482.005

AR target gene expression. Further analyses took into account the effect of loss of coregulator expression on the magnitude of androgen regulation of AR target gene expression. Genes for which androgen regulation was decreased by specific coregulator knock-down as compared to control transfection were scored as negative; those for which androgen regulation increased upon coregulator loss were scored as positive (*Supplementary file 5*). Strikingly, each coregulator studied could simultaneously increase the androgen-responsiveness of a subset of AR target genes under its control (thus acting as a coactivator) and decrease androgen-responsiveness of other genes that rely on it for androgen regulation (acting as corepressor) (*Figure 1B*). These results reveal previously unrecognized context-dependency in the manner by which coregulators govern androgen regulation of gene expression.

Next, the effect of loss of coregulator expression on the direction of androgen regulation of gene expression was analyzed. AR target genes for which loss of coregulator expression altered the absolute level but not the direction of androgen regulation (e.g., upregulated by androgens in both control and knock-down conditions) were considered as having consistent directionality whereas those in which silencing coregulator expression changed both the magnitude and the direction of androgen regulation (e.g., from up- to down-regulated) were considered as exhibiting 'opposite' directionality (*Supplementary file 5*). The vast majority of genes fell into the consistent category (*Figure 1B*).

Androgen-responsiveness of the same AR target gene could be affected by multiple coregulators. 2 to 4 coregulators affected the androgen-responsiveness of the vast majority of genes. For a few genes only, that number of coregulators ranged from 0 to 14. For instance, androgen-responsiveness of the genes encoding *GNB4* and *RAB27A* was modified by 4 or 12, respectively, individual coregulators (*Supplementary file 5*). ChIP studies using antibodies directed against 6 representative coregulators verified the correlation between androgen-dependent recruitment of NCOA3, SMARCA4 or WDR77 to AREs and the pattern of androgen-responsiveness of *GNB4* and *RAB27A* in

LNCaP cells. Conversely, coregulators which loss did not affect the androgen regulation of these genes, such as PKN1 or NCOA2, were not found at AREs of these genes. Yet other coregulators (e.g. EP300) were present at relatively high basal level at these AREs, but modification of the androgen regulation of the corresponding gene was not noted unless there were marked changes in recruitment of the coregulator to those AREs (for instance *RAB27A*) (*Supplementary file 5*).

Time course studies were performed to determine the kinetics of representative coregulator recruitment to AREs in these genes. At 1 hr, 4 hr, 16 hr, and 48 hr after treatment, cells were harvested for ChIP analysis of WDR77, NCOA3, and AR. Robust androgen-induced recruitment of AR was seen after 1 hr, 4 hr, or 16 hr at AREs in both genes, which became less pronounced (RAB27A) or not detectable (GNB4) at 48 hr. Androgen-stimulated binding of WDR77 and NCOA3 occurred at all time points at AREs of both genes (*Supplementary file 5*). The kinetics by which androgen regulation of GNB4 and RAB27A is increased after siRNA-mediated silencing of WDR77 and NCOA3 was defined in real-time RT-PCR studies in LNCaP cells. Consistent with a lag between recruitment of AR and changes in androgen responsiveness of its target genes (*Massie et al., 2011*), androgen stimulation of GNB4 and RAB27A was first seen at 4 hr. At 16 hr androgen treatment, silencing of both WDR77 and NCOA3 increased the level of androgen regulation of GNB4 as well as RAB27A. For both coregulators and both genes studied, this effect was more pronounced at 48 hr. The siRNA-mediated decrease in WDR77 and NCOA3 expression respectively, however, was similar after 1 hr, 4 hr, 16 hr, or 48 hr of treatment (*Supplementary file 5*). These results indicated no marked differences in the kinetics of coregulator recruitment to AREs in these target genes.

The possibility that multiple coregulators work in concert to control androgen regulation of individual target genes was examined further. Pairwise comparison between all coregulator-dependent AR target gene sets was done to determine the overlap in number of genes. With few exceptions (e.g. EP300-BAG1) the overlap in genes between different coregulator-dependent signatures was less than 30%. Among the 136 pair-wise comparisons, only 42 (30.8%) and 31 (22.7%) have significant overlap at the level of $p<0.05$ and $p<0.01$, respectively (*Figure 1—source data 2*). For instance, between STAT3- and WDR77-dependent gene sets, 30 genes overlapped, which corresponds to 31% and 28% of gene signatures, respectively (p=0.99 for significance of this overlap, *Figure 1C*). Of note, the directionality in androgen regulation was preserved for only 15 genes (or 16% and 14%) (*Figure 1C*, insert).

Because of these findings, we analyzed whether AR target genes could be grouped based on the degree to which their androgen-responsiveness depends on specific coregulators. For this analysis, effect of coregulator loss on magnitude of androgen regulation, but not its direction, was considered. RCHY1 knock-down results were not included as loss of this coregulator did not affect androgen-regulation of any of the 452 AR target genes. Relatively small groups of AR target genes for which androgen-dependency was either decreased or increased clustered together (*Figure 1D*). Mutual exclusivity in coregulator dependency of androgen regulation was noted for some AR target gene groups (blue line, EP300), and androgen-responsiveness of other gene sets was affected by 2 or more coregulators (e.g. orange line, EP300 and SMARCA4). This low level of coregulator cooperativity was supported further by Pearson correlation analyses (*Figure 2—figure supplement 1*). Findings of striking AR target gene preference and context-dependence among coregulators indicated that specific coregulator dependency can differentiate between molecular modes of androgen action in CaP.

## Molecular determinants, biological and clinical relevance

Since kinetics studies (*Supplementary file 5*) did not reveal marked differences in the timing of coregulator recruitment to AREs, the possibility that the composition of ARBSs associated with individual coregulator-dependent AR target gene signatures differs was examined. DNA sequences corresponding to ARBSs were retrieved, expanded by 1 kb at the 5' and 3' ends and analyzed using Cistrome Project tools (*Liu et al., 2011*). A total of 283 significantly overrepresented TF binding motifs were identified (*Figure 2—source data 3*). The number of overrepresented motifs in ARBSs of an individual gene signatures ranged from 4 to 127 (*Figure 2A*) and did not correlate with the number of genes per coregulator-dependent gene signature or corresponding number of ARBSs (some genes harbor >1 ARBS). The predominant motif overrepresented in ARBSs from the 452 gene list and the 17 subgroups was one that matched the binding site for AR. ARBSs that did not contain an overrepresented ARE harbored a related, interchangeable motif such as that recognized by

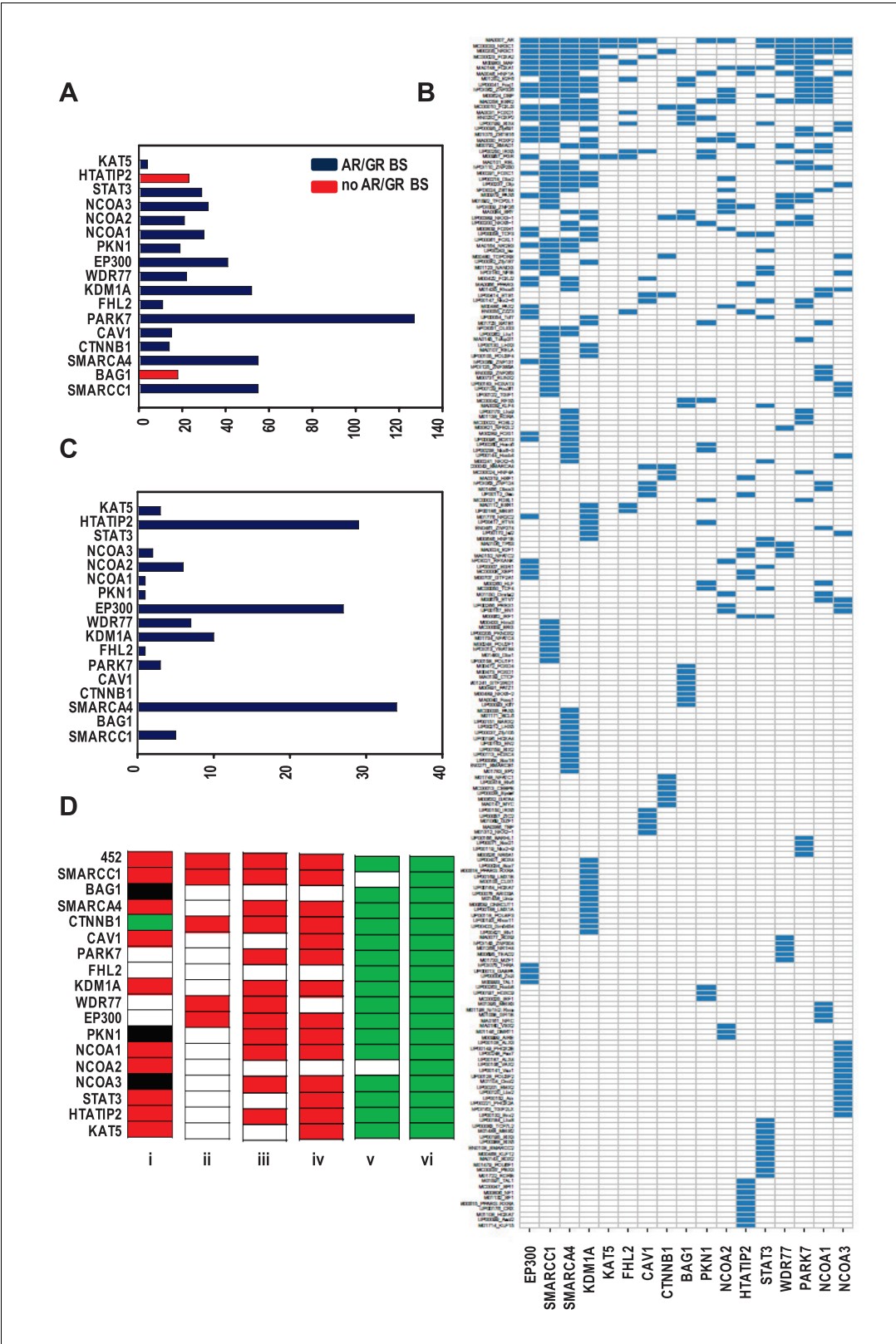

**Figure 2.** Organization of ARBSs, cell biology and clinical relevance associated with the coregulator-dependent AR target gene signatures. (**A**) Number of overrepresented TF binding sites (TFBSs), and AR or GR binding sites (BSs) that are identified in ARBSs of coregulator-dependent AR target gene sets using Cistrome project tools. *Figure 2—source data 1* provides numerical information for data shown here. (**B**) Heatmap summarizing clustering of overrepresented TFBSs within and among ARBSs in the AR target gene sets. Blue bar, one overrepresented TFBS. (**C**) Number of IPA categories that

*Figure 2 continued on next page*

*Figure 2 continued*

associate with individual coregulator-dependent AR target gene signatures. *Figure 2—source data 2* provides numerical information for data shown here. (D) GSEA analyses of coregulator-dependent AR target gene sets between normal prostate from patients treated with dutasteride versus vehicle treatment (GSE9972) (i), between localized CaP versus normal prostate (GSE21034) (ii), between localized CaP that recurs biochemically versus localized CaP that does not recur biochemically (GSE21034) (iii), between localized versus metastatic CaP (GSE32269) (iv), between CaP versus normal bone marrow (GSE32269) (v) and between luminal versus basal epithelial prostate cells (GSE67070) (vi). Red, significant negative enrichment; green, significant positive enrichment; white, no enrichment; black, NES could not be determined. False discovery rate <0.25 was considered significant enrichment.
DOI: https://doi.org/10.7554/eLife.28482.008

The following source data and figure supplement are available for figure 2:

**Source data 1.** Overview of number of TF binding sites (TFBSs), and AR or GR binding sites (BSs) that are identified in ARBSs of coregulator-dependent AR target gene sets using Cistrome project tools.
DOI: https://doi.org/10.7554/eLife.28482.010

**Source data 2.** Overview of the number of Ingenuity Pathway Analysis categories that associate with individual coregulator-dependent AR target gene signatures.
DOI: https://doi.org/10.7554/eLife.28482.011

**Source data 3.** Overview of transcription factor (TF) binding sites identified in ARBSs present in 452 AR target genes.
DOI: https://doi.org/10.7554/eLife.28482.012

**Figure supplement 1.** Correlation of coregulator-dependent androgen-responsiveness of AR target genes.
DOI: https://doi.org/10.7554/eLife.28482.009

glucocorticoid receptor (GR) (e.g., BAG1- and HTATIP2-dependent genes). Motifs known to be enriched in close proximity to an ARE, such as Forkhead family members, were overrepresented in subgroups except for those in NCOA3-dependent AR target genes. Consensus binding sites for other TFs were shared by ARBSs that are present in multiple AR target gene sets. Strikingly, however, each individual coregulator-dependent AR target gene set harbored in its ARBSs at least one overrepresented TF binding motif that was not found to be overrepresented in ARBSs in other AR target gene lists or in the overarching 452 gene set (*Figure 2—source data 3*). For instance, multiple TF binding sites that are enriched selectively in STAT3-dependent gene ARBSs but not in other subsignatures correspond to TFs that function in development, organogenesis and stemness (e.g MEIS2, NANOG, SOX2) (*Figure 2—source data 3*). This recalls the emerging role for STAT3 and AR in cancer stemness (*Li et al., 2015*; *Kregel et al., 2013*; *Kregel et al., 2014*) and the overlap in NANOG binding sites with a subset of ARBSs (*Jeter et al., 2016*). Unsupervised clustering confirmed gene set-specific TF binding site clustering in ARBSs (*Figure 2B*), and suggested that selective motif enrichments serve as the molecular basis for coregulator-dependent clustering of AR target genes. Despite the overall low correlation between coregulator-dependence of AR target genes (*Figure 2—figure supplement 1*), we obtained 4 combined gene sets by combining the 4 pairs of gene sets with highest Pearson correlation (≥0.25) and identified significantly overrepresented TF motifs in each of them. The results of analyses for these individual or combined gene sets (*Figure 2—figure supplement 1*) confirm the observation of selective enrichment of TFBSs.

The possibility that individual coregulator-dependent AR target gene sets control different aspects of androgen-dependent CaP cell biology was explored. Using Ingenuity Pathway Analysis (IPA), we determined the association of each of these individual gene signatures with biological functions. The entire 452 AR target gene set was associated with 36 categories, whereas individual AR target gene signatures were associated with 0 (e.g. CAV1) to 34 (for SMARCA4) categories (*Figure 2C*, *Supplementary file 3*). Also, contrary to the wide range of biological processes associated with the entire 452-gene set, coregulator-dependent AR target gene sets tended to involve specific biological processes (*Supplementary file 3*). For instance, SMARCC1-dependent AR target genes associated significantly with functions involved in cell death and survival. Similarly-sized gene sets such as those associated with SMARCA4 (n = 124) and KAT5 (n = 142) were associated with a widely different number of categories (34 and 3, respectively). The top 10 IPA canonical pathways most significantly associated with each gene set also markedly differed in composition (*Supplementary file 3*). These results indicate that coregulators may contribute selectively to specific androgen-dependent biological processes that make up the androgen response of CaP cells. IPA analysis on combined or individual gene sets with highest correlation confirmed conclusions of selective enrichment (*Figure 2—figure supplement 1*).

The CaP-specificity and clinical relevance of different coregulator-dependent AR target gene signatures was evaluated using prostate and CaP gene expression profiles that are available in the public domain (*Figure 2D*). First, the possibility that androgen regulation of the coregulator-dependent gene sets differ between benign prostate and CaP was studied. mRNA expression profiles derived from microdissected benign prostate epithelial cells from patients who were either treated with dutasteride, a dual SRD5A inhibitor that prevents conversion of testosterone to the most bioactive androgen dihydrotestosterone, or vehicle prior to radical prostatectomy were compared using gene set enrichment analysis (GSEA) (*Subramanian et al., 2005*). Significant enrichment was determined by FDR q-value. The expression of the overarching 452 gene signature was negatively enriched in benign cells from patients who received dutasteride, indicating androgen regulation of this gene set in normal benign prostate epithelial cells. Similar negative enrichment was found for the majority of evaluable coregulator-dependent subsignatures. For 4 gene subsignatures (PARK7-, FHL2-, WDR77-, and EP300-dependent), however, no changes were seen between dutasteride-treated versus vehicle-treated patients, indicating no androgen regulation of these genes in normal benign epithelial prostate cells. One signature, CTNNB1, was slightly enriched in dutasteride-treated patients. Second, mRNA expression profiles derived from localized CaP and benign prostate were compared. Consistent with previous observations (*Tomlins et al., 2007*; *Heemers et al., 2011*), expression of the 452 gene signature was significantly and negatively enriched in CaP compared with benign prostate. Analyses for the 17 coregulator-dependent AR target gene sets indicated significant negative enrichment for 4 signatures. Third, GSEA was done on CaP gene expression profiles from patients who experienced biochemical failure versus those who did not and on tissues from patients with localized CaP versus metastatic CaP. In each study, the global 452 AR target gene set had a significant NES in the most aggressive state (biochemically recurring or metastatic CaP) but one or more of the 17 coregulator-dependent subsignatures showed no enrichment between recurring and non-recurring CaPs or between localized CaP or CaP that had spread. As AR controls a transcriptional program in normal prostate, which could cloud the assessment of cancer specificity of the gene sets studied, GSEA was done also using gene expression profiles from CaP and normal bone marrow, the most common site of metastatic CaP seeding. This comparison confirmed significant positive enrichment in cancer for all but 2 AR target gene sets (SMARCA4- and NCOA2-dependent). Finally, GSEA using profiles from luminal and basal prostate epithelial cells verified luminal origin of the signatures studied here. Importantly, the size of the gene lists did not correlate with the NES significance. These findings suggest differential involvement of select coregulator-dependent AR target gene sets in initiation and progression of CaP and validate CaP-specificity and luminal association of the signatures studied.

## Novel AR-WDR77-p53 transcriptional code

The results above pointed to the existence of discrete coregulator-dependent mechanisms of AR action that may control select aspects of CaP cell biology, differ in clinical relevance, and be governed by specific coregulator-AR-TF interactions (*Figure 3A*). An integrated review of results for the 17 individual AR target gene sets was done in search for evidence for such transcriptional codes. First, data from Cistrome studies were mined to isolate TF binding sites that were enriched selectively in ARBSs of no more than 2 coregulator-dependent AR target gene signatures (*Figure 2— source data 3*). Next, TFs predicted to bind to these motifs were prioritized based on their significance to clinical CaP progression and functional relevance to CaP cell biology. p53, a well-known tumor suppressor with TF function that regulates apoptosis, cell senescence, cell cycle progression, and DNA repair, and undergoes gain-of-function mutations during CaP progression (*Robinson et al., 2015*; *Hong et al., 2015*), best fit those selection criteria (*Figure 2—source data 3*). Consensus binding motifs for p53 were enriched selectively in ARBSs of WDR77-dependent AR target genes. Five (cell death and DNA replication, recombination and repair) of 7 IPA-identified biological processes that associated significantly with the WDR77-dependent AR target genes were very consistent with p53 function (*Supplementary file 3*). WDR77-dependent AR target genes included for instance *MYC*, a well-known p53 target gene with pivotal roles in CaP aggressiveness and progression. In addition, IPA identified p53 as an upstream regulator function of WDR77- and AR-regulated genes (p=1.37.e$^{-7}$).

That p53 and WDR77 may cooperate to regulate CaP cell response to androgens was examined first in Co-IP studies. Despite previous reports of functional interaction between AR and p53

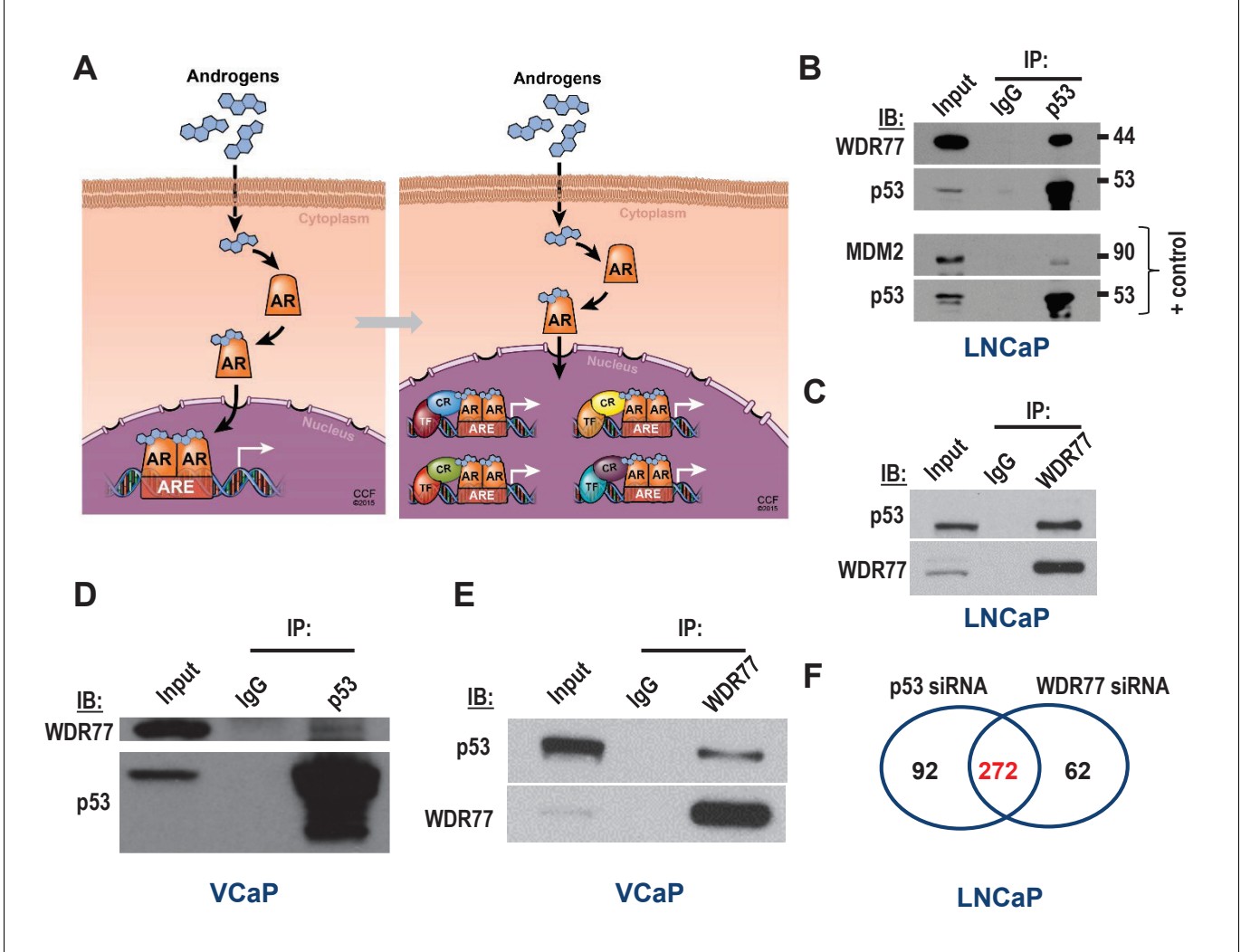

**Figure 3.** Isolation of coregulator-dependent AR transcriptional codes. (**A**) Global AR action in CaP cells partitions into discrete transcriptional mechanisms in which ARE-bound AR interacts with select coregulators (CRs) and transcription factors (TFs) to control distinct biological processes and CaP aggressiveness. (**B, C**) Co-immunoprecipitation assays in which order of antibody used for co-immunoprecipitation and immunoblotting is switched show interaction between WDR77 and p53. Co-IP is done in LNCaP cells that express wild-type p53. Interaction between p53 and MDM2 is shown as a positive control. IP, immunoprecipitation; IB, immunoblotting. (**D, E**) Co-immunoprecipitation assays in which order of antibody used for co-immunoprecipitation and immunoblotting are switched show interaction between WDR77 and p53 in VCaP cells that express p53 R248Q. (**F**) Overlap in genome-wide p53- and WDR77-dependent androgen-responsive gene expression. Results reflect the effect of 48 hr treatment of LNCaP cells with 5nM R1881. R1881 or vehicle treatment was administered 42 hr after siRNA transfection. Numbers, number of genes.
DOI: https://doi.org/10.7554/eLife.28482.013

(*Dean and Knudsen, 2013*; *Guseva et al., 2012*; *Cronauer et al., 2004*; *Shenk et al., 2001*; *Gurova et al., 2002*), these proteins have never been found to interact directly. However, interaction between WDR77 and AR has been described (*Li et al., 2013*). Therefore, the possibility that p53 and WDR77 interact physically was explored. Co-immunoprecipitation assays were done in both directions (i.e. immunoprecipitation for p53 and immunoblotting for WDR77, and immunoprecipitation for WDR77 followed by western blotting for p53) in LNCaP cells, which express wild-type p53. These studies revealed that WDR77 and p53 are part of the same immune complex (*Figure 3B and C*). These same experiments were performed in VCaP cells, which confirmed the presence of p53 and WDR77 in the same protein complex regardless of the order in which the immunoprecipitation or -blotting was done (*Figure 3D and E*). Next, the overlap in p53- and WDR77-dependence of androgen-responsive gene expression was defined. LNCaP cells were transfected using siRNA

targeting p53 or WDR77, or non-targeting control siRNA. Cells were then treated with R1881 or vehicle as above. Illumina genome-wide HTv4 BeadChip analyses were done, and genes that relied on p53 or WDR77 for androgen-responsiveness were identified as above. As shown in *Figure 3F*, androgen regulation of a set of 272 genes was affected similarly by knockdown of p53 or WDR77 (p<2.2E-16). Remarkable consistency was noted in the impact on directionality of androgen-regulation of these genes after silencing of p53 or WDR77. 76.9% of genes for which androgen-responsiveness was increased after p53 knock-down overlapped with genes for which level of androgen-regulation was enhanced upon silencing of WDR77. Conversely, 88.2% of genes for which androgen-responsiveness is enhanced after loss of WDR77 showed an increased level of androgen regulation also in p53 siRNA-transfected cells. Similarly, 68.3% of genes for which androgen-responsiveness decreased after loss of p53 overlapped with genes for which level of androgen-regulation was diminished upon silencing of WDR77; and 67% of genes for which level of androgen-responsiveness was lessened after loss of WDR77 showed decreased androgen regulation also in p53 siRNA transfection condition. For none of the androgen-regulated genes isolated, inconsistency in the directionality of androgen regulation was observed between p53 and WDR77 silencing. These results strongly supported co-operativity between WDR77 and p53 in androgen-regulation of select AR-dependent genes. The ability of cistrome data to predict TF-coregulator interactions was validated by co-immuoprecipitation of STAT3 and IRF1, for which binding sites are enriched selectively in the ARBSs of STAT3- dependent genes (*Supplementary file 6*). Gene expression studies following siRNA-mediated silencing of IRF1 and STAT3 confirmed considerable (n = 413) overlap in androgen-responsive genes (p<2.2E-16) (*Supplementary file 6*). The directionality of androgen regulation of affected genes was preserved in IRF1- or STAT3-knockdown condition. Comparison of IPA results (*Supplementary file 3*) from STAT3-IRF1 and p53-WDR77 interactions indicated both shared and unique molecular functions (*Supplementary file 6*, p<2.2E-16).

## p53-WDR77 interaction during CaP progression

As AR and p53 have been proposed as drivers of lethal CaP (*Robinson et al., 2015*; *Hong et al., 2015*), the relevance of WDR77-p53 interaction for late stage disease was studied further. Recent NextGen sequencing studies have identified p53 mutants that are enriched in castration-recurrent (CR-)CaP, but the contribution of these p53 mutants to AR-dependent transcription is poorly understood (*Cronauer et al., 2004*; *Shenk et al., 2001*; *Gurova et al., 2002*). WDR77 and p53 were part of the same immunocomplex in VCaP cells (*Figure 3D–E*) that endogenously express gain-of-function p53 R248Q. We therefore examined the effect of CaP-specific p53 mutants on androgen regulation of representative WDR77-dependent AR target genes in LNCaP cells via co-expression of 9 clinically relevant CaP p53 mutants (*Figure 4A*). These 9 p53 mutants have been detected recently in tissue and blood from CR-CaP patients (*Hong et al., 2015*). Higher nuclear expression level of p53 mutants than wild-type p53 was observed, which is reminiscent of observations in patient specimens (*Haffner et al., 2013*). Increased p53 nuclear expression did not impact on nuclear content of AR or WDR77 (*Figure 4B*). Despite some heterogeneity in the contribution of different p53 mutants, overall androgen regulation of *GNB4* and *RAB27A* was maintained in the presence of added mutant p53 (modeling clinically relevant heterotetramerization between wild-type and mutant p53 (e.g. [*Muller and Vousden, 2014*])), compared to empty vector or wild-type p53 (*Figure 4C*). These findings suggest that activity of the AR-WDR77-p53 transcriptional code is maintained in CR-CaP that expresses mutant p53. This conclusion is consistent also with GSEA analyses using the WDR7-dependent gene expression signature and gene expression profiles from clinical CR-CaP cases that express mutant p53 versus those that express wild-type p53 (*Grasso et al., 2012*). No significant normalized enrichment score was obtained between 2 groups of cases. Co-IP studies in which 2 representative p53 mutants, C135Y and N239T, were expressed in a p53-null LNCaP subline (*Guseva et al., 2012*) confirmed interaction of mutant p53 with WDR77 (*Figure 4D*).

## Mechanism underlying WDR77-dependent AR and p53 interaction

To gain more insight into the action of the novel AR-WDR77-p53 transcriptional code, IP-mass-spectrometry analysis was performed on nuclear fractions from androgen- versus vehicle-treated LNCaP cells. IP experiments in which either p53 or WDR77 antibodies were used independently identified 3 proteins (14-3-3 sigma, hnRNPU and PGAM5) as part of the AR-WDR77-p53 immunocomplex.

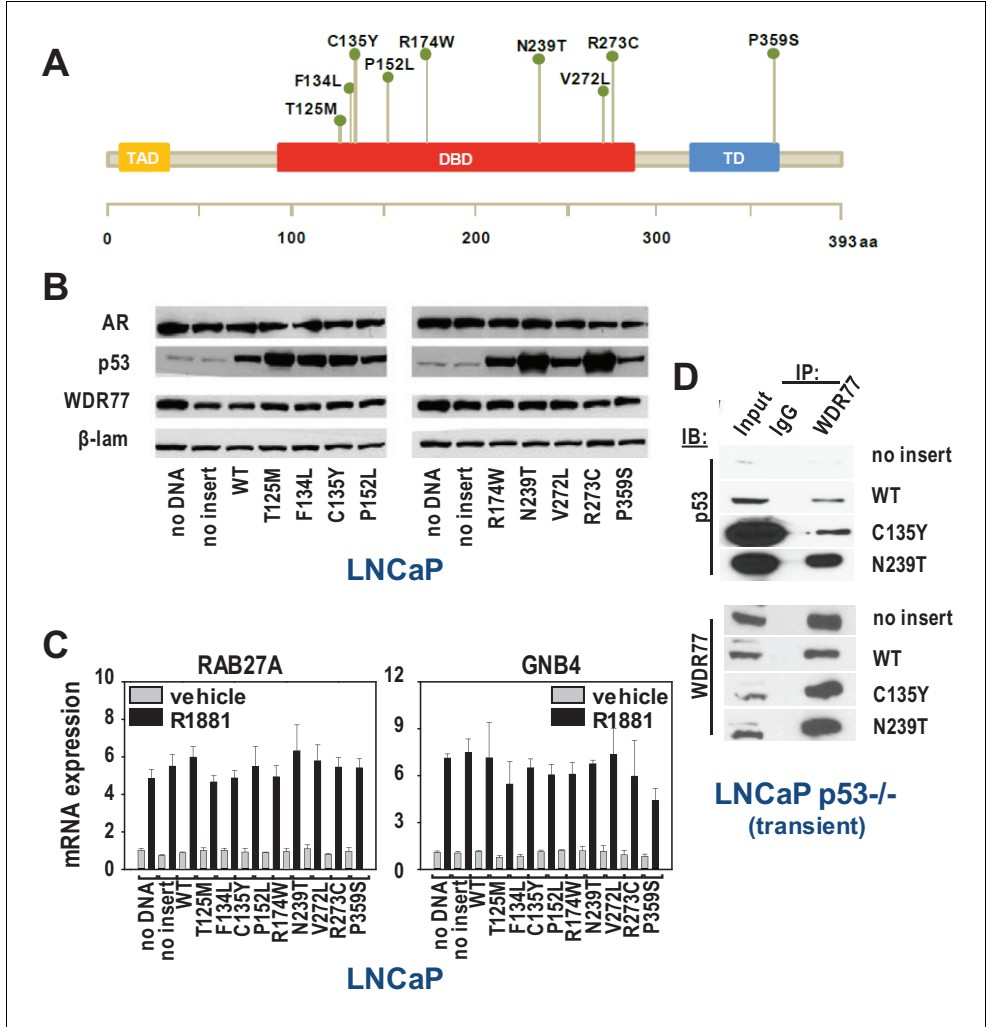

**Figure 4.** WDR77-p53 interaction is maintained in CaP cells that express gain of function p53 mutants. (A) Overview of p53 somatic mutations that are present at the subclonal level in localized CaP and are enriched in metastatic CR-CaP. Mutations studied were reported in (*Hong et al., 2015*). (B) Western blotting on nuclear fractions of LNCaP cells that were transfected with empty vector, vector expressing wild-type p53, or vectors that individually express each of the 9 clinically relevant p53 mutants. To control for loading differences, blots were reprobed for laminin. (C) qRT-PCR analysis of the effect of transfections using empty vector, vector expressing wild-type p53, or vectors that individually express each of the 9 clinically relevant p53 mutants. 16 hr after electroporation (*Schmidt et al., 2012*), LNCaP cells were seeded in medium supplemented with charcoal-stripped FBS. 1 day later, medium was changed and cells were treated with 5nM R1881 or vehicle for 48 hr. RAB27A and GNB4 expression was evaluated using real-time RT-PCR. Target gene mRNA levels were normalized with the values obtained from GAPDH expression and are expressed as relative expression values, taking the value obtained from one of the vehicle-treated samples as 1. Columns, means of values obtained from 3 independent biological replicates; bars, sem. Grey bars, treatment with vehicle; black bars, treatment with R1881. (D) Co-IP studies in LNCaP cells in which p53 expression had been silenced (*Guseva et al., 2012*), and subsequently transfected using empty vector, vector encoding wild-type p53 or p53 mutants C135Y or N239T. Cells were harvested 72 hr after transfection. Transient, LNCaP cells in which endogenous expression of p53 had been silenced (*Guseva et al., 2012*) were transiently transfected with empty vector, or expression vectors for wild-type p53, p53 C135Y or p53 N239T. IP, immunoprecipitation; IB, immunoblotting
DOI: https://doi.org/10.7554/eLife.28482.014

PGAM5, which has previously been reported to play a role in regulation of cell death (*Vaseva et al., 2012*; *Wang et al., 2012*) but was not known to be relevant to AR signaling or CaP biology, was prioritized for validation. First, immunohistochemistry for PGAM5 was performed on tissue microarrays that contain 29 benign prostate and 151 CaP tissues. Expression of PGAM5 was significantly higher in CaP than in benign prostate (score of 2.24 vs 1.58, p=0.00059, t-test) and was higher also in CaPs of Gleason scores 7–10 than in CaP of Gleason score 6 (p<0.05, t-test) (*Figure 5A,B*). These findings were in line with queries of the Oncomine and cBioPortal databases which indicated also overexpression of PGAM5 in CaP versus benign prostate, and increased PGAM5 expression with increasing Gleason grade and CaP progression (*Figure 5C*). These results indicated that PGAM5, as other AR-associated coregulators studied here, is overexpressed in CaP where it correlates with more aggressive CaP behavior. The implications of PGAM5 function for WDR77-dependent AR action were determined next. PGAM5 exists in 2 isoforms (long and short). Co-IPs using antibodies directed against AR or p53 for IP confirmed the presence of both PGAM5 forms in the AR-WDR77-p53 complex in LNCaP cells (*Figure 5D*). Androgen treatment induced recruitment of PGAM5 as well as p53 to ARE-containing regions within the genes encoding *GNB4* and *RAB27A*, to which WDR77 and AR also bind (*Figure 5E,F*). Knock-down of PGAM5 mirrored the effect of WDR77 loss on androgen regulation of target genes such as *RAB27A* and *GNB4*. Genome-wide oligoarray expression profiling in LNCaP cells showed that loss of PGAM5 altered the androgen responsiveness of 218/272 of the p53 and WDR77-dependent genes (p<2.2E-16) (*Figure 5G*), and for each of those genes the impact on directionality of androgen regulation was the same for PGAM5 loss as that observed after silencing of WDR77 and p53. At the molecular level, loss of WDR77 altered androgen-dependent pattern of coimmunoprecipitation of p53 by AR (*Figure 5H*), and silencing of either WDR77 or PGAM5 prevented androgen-dependent recruitment of p53 to WDR77-dependent target genes (*Figure 5I*). Moreover, ChIP-re-ChIP experiments using an antibody targeting p53 for ChIP and an antibody against AR for Re-ChIP confirmed androgen-stimulated co-recruitment of AR and p53 to these genes, which was decreased following siRNA-mediated silencing of WDR77. Similar results were obtained in parallel experiments in which ChIP targeted AR and Re-ChIP was directed at p53 (*Figure 5I*). In combination, these data demonstrate a novel mechanism in which AR-associated coregulators WDR77 and PGAM5 control androgen-dependent recruitment of p53 to ARBSs in a subset of AR target genes.

## Biological consequences of AR-WDR77-p53 interaction

We set out to verify the IPA results of the WDR77-, p53- and PGAM5-dependent androgen-responsive gene signature, which indicated preferential roles in regulation of cell survival, cell death and cell proliferation (*Supplementary file 3*). Since the regulatory proteins WDR77 and PGAM5 may be differentially involved in other transcriptional complexes, which could confound interpretation of results, 4 representative WDR77-dependent AR target genes, *GNB4* and *RAB27A* as well as *HES6* and *AGR2* (previously reported as ARE-driven genes and identified in our 452 gene-signature also) (*Sharma et al., 2013*; *Ramos-Montoya et al., 2014*) were included in these analyses. siRNA-mediated silencing of all regulators and target genes markedly reduced cell viability of LNCaP cells, both under normal culture conditions and under androgen-stimulated conditions (*Figure 6A,B*). Similar effects were observed when experiments were repeated in p53-null LNCaP sublines that are stably transfected with expression constructs encoding p53 mutants C135Y and N239T or in VCaP cells that express p53 R248Q (*Figure 6C,D*). Except for GNB4, for which effects were modest, propidium iodide FACS studies in LNCaP cells indicated that this reduced cell viability may be due to slow G1 phase progression and G1/S transition (*Figure 6E*). Western blot analysis on parallel samples showed marked increases in expression of gamma-pH2AX, supporting potential effects on cell cycle stage, early apoptosis and/or DNA damage (*Figure 6F*).

## Discussion

This first systematic analysis of the individual contribution of multiple coregulators to androgen regulation of several hundred *bona fide* AR target genes has revealed remarkable diversity in the molecular modulation of AR-dependent transcription. By integrating results from diverse analyses, important novel insights in AR action in CaP, including 2 novel AR-dependent transcriptional codes

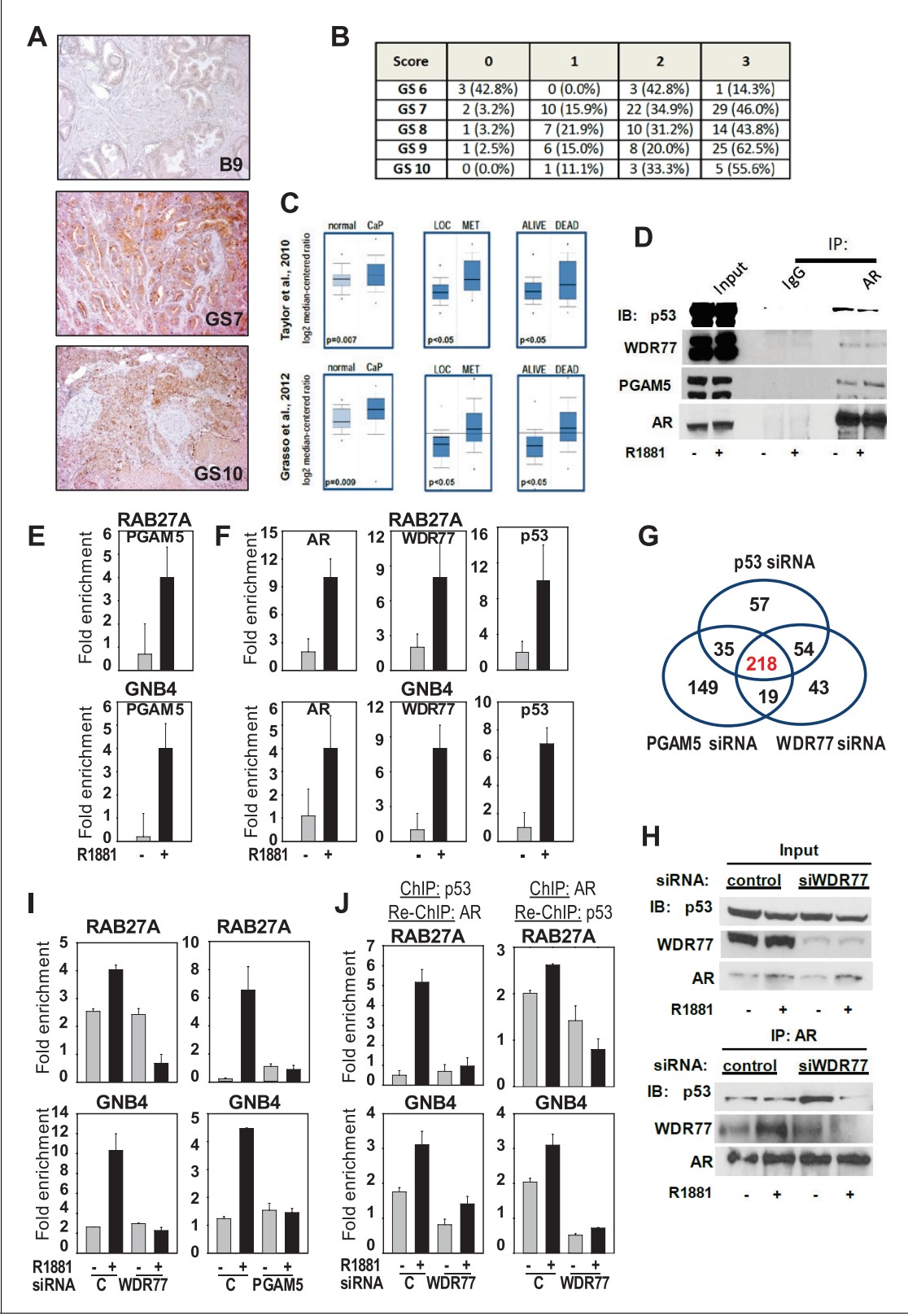

**Figure 5.** Mechanism underlying WDR77-dependent AR and p53 interaction. (**A**) Representative images of prostate and CaP TMA cores that were subjected to immunohistochemistry using an antibody directed against PGAM5. B9, benign prostate; GS, Gleason score. 10x magnification. (**B**) Overview of quantitation of PGAM5 immunohistochemistry on CaP cores. Columns represent different scores. Rows represent CaP cores that are grouped according to Gleason scores. GS, Gleason score. Number, number of TMA cores; %, percentage of cores per GS group. (**C**) Oncomine
*Figure 5 continued on next page*

*Figure 5 continued*

analyses done on 2 CaP gene expression profiling studies. PGAM5 mRNA expression was determined in normal prostate versus CaP (left panels), in localized (LOC) CaP versus metastatic (MET) CaP (middle panels), and taken into consideration follow-up information on 5 year survival status of patients (alive or dead). *Figure 5—source data 1* provides more information on PGAM5 peptides identified using IP-mass spectrometry. (D) Co-immunoprecipitation assay shows interaction between AR, WDR77, PGAM5 and p53. Co-IP is done in LNCaP cells after 16 hr treatment with R1881 or vehicle. IP, immunoprecipitation; IB, immunoblotting. (E) ChIP validation of androgen-dependent recruitment of PGAM5 to AREs within WDR77-responsive AR target genes RAB27A and GNB4. Results reflect the effect of 16 hr treatment of LNCaP cells with 5nM R1881 or vehicle. (F) ChIP validation of androgen-dependent recruitment of AR, PGAM5 and p53 to the same AREs within WDR77-responsive AR target genes RAB27A and GNB4. Culture conditions are as under E. (G) Overlap in genome-wide p53-, WDR77-, and pGAM5-dependent androgen-responsive gene expression. Results reflect the effect of 48 hr treatment of LNCaP cells with 5nM R1881. R1881 or vehicle treatment was administered 42 hr after siRNA transfection. Numbers, number of genes. (H) Co-immunoprecipitation assay demonstrates WDR77-dependence of AR and p53 interaction. Co-IP is done in LNCaP cells after 16 hr treatment with R1881 or vehicle. Treatment was given 72 hr after transfection using siRNAs targeting WDR77 or control siRNAs. (I) ChIP validation of reliance of androgen-dependent recruitment of p53 to WDR77-responsive AREs on WDR77 and PGAM5. Culture conditions are as under E. (J) ChIP-re-ChIP experiments validate androgen- and WDR77-dependent co-recruitment of p53 and AR to the same GNB4 and RAB27A gene regions. Culture conditions and data representation is as under H. Data shown in panels D-I are derived from LNCaP cells.

DOI: https://doi.org/10.7554/eLife.28482.015

The following source data is available for figure 5:

**Source data 1.** PGAM5 peptides identified after IP-mass spectrometry.

DOI: https://doi.org/10.7554/eLife.28482.016

(AR-WDR77-p53 and AR-STAT3-IRF1) and a novel AR-associated coregulator (PGAM5), have been derived. These findings strengthen the concept and feasibility of selective ADT.

Previous efforts to target for therapy the heterogeneity in AR action have focused on enhancing AR action in a tissue-specific manner via development of selective AR modulators (*Pihlajamaa et al., 2015*). The alternative, namely blocking AR action in specific tissues while not affecting it in organs where its sustained activity is required for maintenance of normal function, has not yet been attempted. We reasoned that a better understanding of the contribution of coregulators, the master regulators for nuclear receptor (NR)-mediated transcription, to ARE-driven gene expression in CaP would facilitate such a CaP-selective approach to treatment. The studies described above demonstrate that AR action in CaP can be broken down in coregulator-dependent fractions. Of note, results were derived from the same cell type, in which AR is activated by the same non-metabolizable ligand that is administered using a standardized timing and dosing scheme, and relies on the same endogenously expressed cell-type specific array of coregulators to execute AR-dependent transcription. Our work has thus isolated a previously unrecognized intracellular level of heterogeneity in AR action that differentially uses critical regulators of transcriptional machinery to induce AR target gene subsets.

The finding that androgen action depends on different coregulation of transcription is consistent with previous studies that mostly involved silencing of one coregulator and a handful ARE-driven gene fragments or global androgen-dependent gene expression patterns (*Marshall et al., 2003*; *Ianculescu et al., 2012*; *Xu et al., 2009*; *Heemers et al., 2009*). The scope and nature of our approach revealed, for the first time, patterns of (co)regulation of AR action in CaP cells and its molecular underpinning and biological relevance. Androgen regulation of most AR target genes could be affected by more than 1 coregulator, but the overlap in different contributing coregulators between target genes was limited. When AR target genes were grouped based on androgen-responsiveness to coregulators, striking differences in ARBS TF binding site composition were found between the AR target gene subsignatures. These findings are reminiscent of a previously proposed model in which DNA not only serves as a binding site for TFs, but also as an allosteric ligand for DNA-bound NRs (*Meijsing et al., 2009*). In this model, the DNA sequence at the NR binding site controls the composition of transcriptional complexes formed at that site. This fits also with the concept of the Androgen Response Unit (ARU) in which variability in ARE-driven transcription was attributed to sequence composition of the regions close to AREs to which other proteins bind and cooperate with AR (*Robins et al., 1994*). Our methodical documentation of the extent of this variability, and the identification of select candidate contributing TF binding sites, provide the first glimpses of differential composition of AR transcriptional complexes at individual target genes. The proposed transcriptional codes and the model of cooperative or mutually exclusive coregulator-

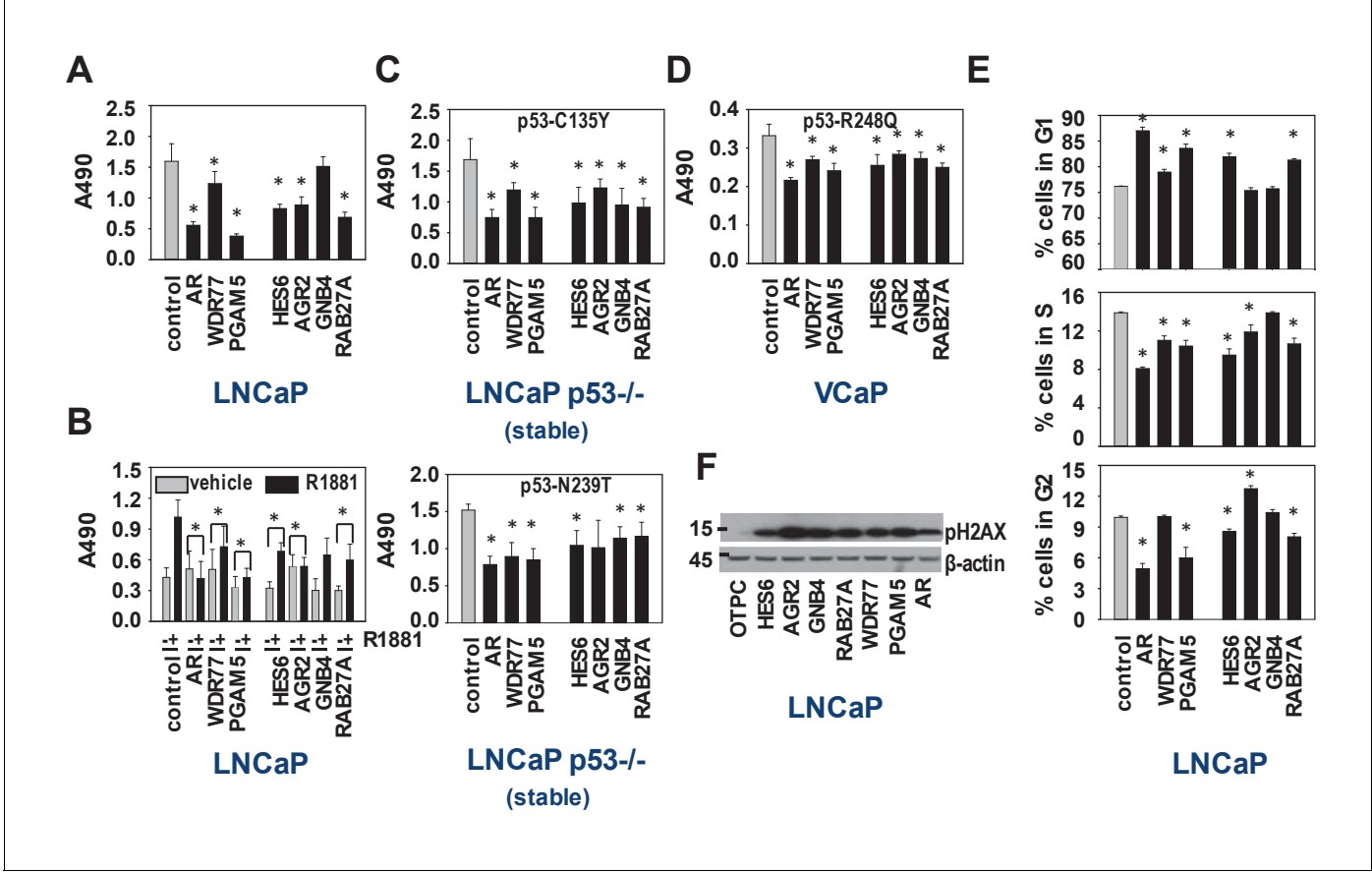

**Figure 6.** Biological consequences of WDR77-dependent AR and p53 interaction. (**A**) Silencing of regulators and effectors of the AR-WDR77-p53 transcriptional code decreases CaP cell viability. LNCaP cells were transfected with siRNAs targeting AR, WDR77, PGAM5, HES6, AGR2, GNB4 or RAB27A, or control siRNAs. Medium was changed 16 hr later, and an MTS assay reading absorbance at 490 nm was performed 96 hr after transfection. Columns, Mean values from five individual measurements; bars, SEM. (**B**) Viability experiment as under (**A**) except that cells were treated for 96 hr with vehicle (grey bars) or 5nM R1881 (black bars) at medium change after 16 hr. (**C**) Viability experiment described under (**A**) in p53-null LNCaP cells that stably express p53 C135Y (top panel) or N239T (bottom panel). (**D**) Viability experiment described under (**A**) in VCaP cells that endogenously express p53 R248Q. (**E**) Silencing of regulators and effectors of the AR-WDR77-p53 transcriptional code delays cell cycle progression. LNCaP cells were transfected and harvested as above. Cells were stained using propidium iodide and flow cytometry was performed. Columns, Mean values from three individual biological replicates; bars, SEM. (**F**) Silencing of regulators and effectors of the AR-WDR77-p53 transcriptional code alters phospho-H2AX expression. LNCaP cells were transfected as above. Cells were harvested and total protein extracts were subjected to western blot analysis using an antibody directed against phospho-H2AX. To control for potential loading differences, blots were stripped and reprobed with an antibody targeted at β-actin. *Figure 6—source data 1* provides p-values for the data shown in panels A, B, C, D and E. *, significance with p<0.05 compared to control siRNA condition; stable, LNCaP cells in which endogenous expression of p53 had been silenced were stably transfected with expression vectors for p53 C135Y or p53 N239T.

DOI: https://doi.org/10.7554/eLife.28482.017

The following source data is available for figure 6:

**Source data 1.** Summary of p-values for data presented in *Figure 6*.
DOI: https://doi.org/10.7554/eLife.28482.018

coregulator interactions support the diversity and modularity in the coregulator component of transcriptional complexes (*Malovannaya et al., 2011*). Context-dependency in the contribution of individual coregulators as activator or repressor of AR activity, described for LSD1/KDM1A before (*Cai et al., 2011*), may be related to the presence of multiple functionally diverse isoforms (*Djebali et al., 2012*) of coregulator genes in a target cell, or to specific post-translational modifications on the intracellular pool of a coregulator.

That heterogeneity in coregulator-dependent AR target gene subsignatures is associated with different cell biology processes and clinical CaP progression is novel. This result has important

implications for therapeutic intervention as it lends further credence to the concept of selective, CaP-specific forms of ADT. Theoretically, blocking a CaP-specific segment of AR transcriptional output that controls aggressive CaP cell behavior and clinical progression will lead to CaP remission while bypassing resistance and avoiding side effects of conventional ADT (*Heemers, 2014*; *Nguyen et al., 2015*). The AR-WDR77-p53-dependent transcriptional mechanism highlighted here exemplifies such clinical relevance and therapeutic potential: it mediates cell cycle progression, is androgen-regulated specifically in CaP but not benign prostate, and -unlike other AR target gene signatures- is maintained in CR-CaP. The resulting protein-protein and protein-DNA interactions contribute to AR's control over CaP cell survival, apoptosis and proliferation, which has long been recognized but is poorly understood at the molecular level (*Ta and Gioeli, 2014*; *Wen et al., 2014*). The identification of PGAM5, as novel contributing AR coregulator fits with its role in regulation of cell death (*Vaseva et al., 2012*; *Wang et al., 2012*).

Functional interaction between AR and p53, which have recently been isolated as 2 major genomic drivers of lethal CaP progression (*Robinson et al., 2015*; *Hong et al., 2015*), has been reported, but remains poorly understood (*Dean and Knudsen, 2013*; *Guseva et al., 2012*; *Cronauer et al., 2004*; *Shenk et al., 2001*). Findings that p53 can both down- and upregulate AR-dependent transcription, however, relied on reporter constructs or one AR target gene (typically *PSA*), and, most often, exogenously overexpressed AR and/or p53. Context-dependence such as that observed above for coregulator contribution to AR-mediated transcription may explain these discrepancies. None of these studies detected direct interaction between AR and p53. Interaction between AR and WDR77 has been shown (*Li et al., 2013*) and our studies show that WDR77 mediates recruitment of p53 to AR.

Under ADT, p53 undergoes gain-of-function mutations that affect its TF function and protein interactome, increasing metastatic potential and facilitating CaP cell growth (*Hong et al., 2015*; *Muller and Vousden, 2014*; *Muller and Vousden, 2013*; *Nesslinger et al., 2003*; *Vinall et al., 2006*). p53 mutations are present in >50% of CR-CaP cases. WDR77 is expressed also in clinical CR-CaP (*Peng et al., 2008*), and our data indicate that interaction between WDR77 and mutant p53 is maintained. Deciphering the interactions among WDR77, and wild-type and mutant p53, other components of this AR-dependent transcriptional code, at AR and p53 recruiting genomic sites may, therefore, lead to novel, much needed treatments for CR-CaP. Interfering with a select segment of AR action may be feasible through modulation of coregulator-dependent interaction between AR and secondary TFs that jointly control the expression of ARE-driven gene signatures. Recent developments in peptidomimetics and multivalent peptoid conjugates have allowed for disruption of interaction between AR and some of its coregulators in CaP cell lines and xenografts as well as in ex vivo CaP explants (*Ravindranathan et al., 2013*; *Wang et al., 2016*). The ability of polyamides to prevent binding of AR-interacting TF Oct1 to its genomic binding sites supports the possibility of inhibiting TF-DNA interactions (*Obinata et al., 2016*).

The AR-WDR77-p53 transcription code is one example of an entirely novel molecular mechanism in which coregulator action brings together 2 major clinically relevant drivers of lethal CaP progression to control expression of a subset of AR target genes. Our identification of a STAT3- and IRF1-dependent transcriptional code that differs in biological function from AR-WDR77-p53 collaboration underscores the likelihood for other similar or related (e.g., coregulator-coregulator cooperativity) mechanisms. Yet other models of androgen action may not have been captured because of the selection criteria of our assays. The cut-off of 300 Kb between ARBS and TSS, although sizable, may have prevented identification of genes which form AR transcriptional complexes over longer distances via chromatin looping (*Shang et al., 2002*; *Hsieh et al., 2014*). The requirement of 2-fold androgen regulation may have failed to isolate AR target genes for which androgen regulation is not as pronounced. Our design may have overlooked the contribution of lncRNAs (*Yang et al., 2013*) to AR-dependent transcription. The 18 AR-associated coregulators studied here are only a small fraction of the >270 identified to date (*DePriest et al., 2016*). Selection of coregulators for inclusion relied on knowledge of their differential protein expression in CaP. When these studies were conceived, results of NextGen sequencing efforts using clinical CaP specimens had not been reported, and genomic alterations that could affect coregulator function irrespective of changes in expression level could not be taken into account. A retrospective analysis of 18 coregulators included here was done using data from the 8 NextGen CaP studies on ~1500 clinical CaP specimens that are publicly available through the cBio dataportal (*Cerami et al., 2012*). In combination, somatic mutation and

copy number alterations that affect these 18 genes were present in less than 5% of clinical cases (*Supplementary file 7*). The exception was NCOA2, for which copy number increases fit with over-expression at the protein level as determined by immunohistochemistry.

In conclusion, our alternative approach to define systematically the contribution of coregulators to AR-dependent gene expression in CaP revealed that AR action in CaP partitions according to defined fractions. These segments may be amenable to future development into alternative forms of ADT that inhibit only the most clinically relevant portion of AR action.

## Materials and methods

### Cell culture

LNCaP (RRID:CVCL_1379) and VCaP (RRID:CVCL_2235) cells were obtained from the ATCC and cultured as before (*Heemers et al., 2011*; *Schmidt et al., 2012*). LNCaP cells in which p53 expression is silencing were obtained from the Guseva laboratory (*Guseva et al., 2012*). All cell lines are used for no more than 10 passages. Cells were authenticated by STR profiles and validated further by the consistency of their AR- and androgen-responsiveness. Cells were Mycoplasma-tested every 6 months; all tests were negative.

### Reagents

R1881 was purchased from DuPont (Boston, MA). Antibodies that were used are listed below. siGenome On Target Plus SmartPools were purchased from Thermo-Scientific (Lafayette, CO).

### Western blot analysis

Western blotting was done as described (*Schmidt et al., 2012*).

### siRNA transfection

Cells were seeded, transfected and treated as before (*Schmidt et al., 2012*).

### Real time RT-PCR

RNA isolation, cDNA synthesis and real-time RT-PCR were done as before (*Schmidt et al., 2012*). Primers were synthesized by Integrated DNA Technologies (IDT, Coralville, Iowa). Primer sequences used to quantitate expression of PSA, FN1, SCAP, SERPINB5 and GAPDH have been described (*Heemers et al., 2009*; *Heemers et al., 2007a*; *Heemers et al., 2007b*). Other primer sequences are listed below.

### Cell cycle analysis

Cells were harvested in PBS (Life Technologies, Waltham, MA) and spun down. After resuspension of cells in 200 µl PBS, 2 ml 70% ethanol was added dropwise and cells were kept on ice before adding 50 µl RNAse A (10 mg/ml) and 50 µl propidium iodide (50 µg/ml, Sigma-Aldrich, St.Louis, MO). After incubation at 37C for 30 min, cells were sorted using a Becton Dickinson LSR II flow cytometer. Data was analyzed using ModFit software.

### Cell viability assays

Cell viability was assessed as before (*Schmidt et al., 2012*).

### Co-immunoprecipitation assay

Cells were washed with ice-cold PBS and lysed in cell lysis buffer [20 mM Tris, pH 8.0; 150 mM NaCl; 5 mM MgCl$_2$; 0.5% NP40; 1X EDTA-free protease inhibitor cocktail (Roche)] for 1 hr at 4°C. The protein content of the cell lysates was determined using a Bradford assay. The cell lysate was precleared with 50 µL of lysis-buffer-equilibrated Dynabeads protein G (Life Technologies) or protein G agarose (for *Figure 3B*) for 1 hr at 4°C. 2 mg precleared lysate was incubated with 6 µg of antibody (for p53 IP, use p53 (D01) #sc126, p53 (FL393) #sc6243, p53 (pAB421) #OP03 in equal ratios) at 4°C for overnight. The next day, the antibody–protein complexes were precipitated using Dynabeads (protein G) at 4°C for 3 hr. Immunoprecipitated complexes were washed 4 times with wash buffer [20 mM Tris,

pH 8.0; 150 mM NaCl; 5 mM MgCl$_2$; 1X EDTA-free protease inhibitor cocktail], eluted with 20 µL 2x SDS-PAGE Novex sample buffer, heated at 70°C for 10 min and the supernatant was subjected to western blotting.

## Oligoarray gene expression analyses

RNA was isolated from cells using Trizol (Life Technologies), purified on RNeasy columns (Qiagen, Germantown, MD) and checked for integrity using an Agilent 2100 Bioanalyzer.

## Custom Agilent oligoarray analysis

### Identification of putative AR target genes for inclusion on the Agilent oligoarray

A list of AR binding sites (ARBSs) across the human genome derived from LNCaP cells that had been treated for 4 hr with DHT (100 nM) was downloaded from the Brown laboratory website (http://research4.dfci.harvard.edu/brownlab) (*Wang et al., 2009*). ARBSs within 300 Kb of transcriptional start sites (TSSs) (32,890 locations) were cross-matched with previously published records of genes which had been identified as androgen-regulated genes by microarray studies in LNCaP cells (*DePrimo et al., 2002*; *Nelson et al., 2002*; *Segawa et al., 2002*; *Febbo et al., 2005*; *Velasco et al., 2004*; *Ngan et al., 2009*; *Waghray et al., 2001*; *Xu et al., 2001*). This approach resulted in a list of 900 individual genes, which corresponds to 1590 entries as some genes had multiple TSSs.

### Design of a custom 8×15K Agilent array to assess AR target gene expression

The eArray design features on the Agilent website were used to design a custom 8×15K Agilent expression array. Probes targeting genes of interest were chosen from the Agilent probe catalog using their gene symbol. A linker region was attached to the probes as recommended by the manufacturer. Selected probes were organized in probe groups. Probe groups were used to populate the custom Agilent array. Three major probe groups were generated:

1. Probe group 'AR target genes': 898 genes, 2409 probes. For 898 of the 900 genes that were identified as putative AR target genes by cross-matching tiling array data and gene expression array data, Agilent catalog probes were available.
2. Probe group 'coregulator genes': 180 genes, 541 probes. For 180 of the 181 genes that were identified as AR-associated coregulators by cross-matching tiling array data and gene expression array data, Agilent catalog probes were available.
3. Probe group 'housekeeping genes': 58 genes, 184 probes. This probe group targets genes that are routinely used as housekeeping genes on microarray and real-time RT-PCR array platforms as well as genes for which expression has been shown previously not to change during CaP progression (e.g., [*Nakagawa et al., 2008*]).

As additional controls for overall assay performance, probes targeting genes of non-human origin were included (S. Salar, salmon, 50 probes and N. tobaccum, tobacco, 11 probes). For all probe groups, 4 replicates were included on the array. As per the manufacturer's recommendation, the human expression 50 Agilent replicate probe group was included (10-fold), as was an Agilent control grid (536 features).

Labeling of RNA (One-color), hybridization of the labeled cRNA, washing and scanning of the array slides was done by Roswell Park Cancer Institute's Genomic Shared Resource according to the manufacturer's instructions.

### AR target gene Agilent oligoarray data analysis

Array images were scanned using the G2565AA Microarray scanner (Agilent Technologies) with the gene expression signal extracted using Agilent's Feature Extraction 5.1.1 software. The expression data were then normalized by quantile normalization following by log2 transformation.

The Agilent oligoarray chips were first used to profile RNA samples from LNCaP cells that had been treated with vehicle or androgens (5 nM R1881, 48 hr, 3 replicates each treatment group). An unsupervised hierarchical clustering based on the average linkage of Pearson correlation verified

classification of probes and samples as vehicle- and androgen-treated. The limma program in the Bioconductor package under the R computing environment was used to identify androgen-regulated genes. To control the false positive rate (FDR), the resulting p-values were adjusted using the Benjamini and Hochberg algorithm. A total of 452 androgen-regulated genes were obtained at an FDR cutoff 0.05 plus two-fold change in expression intensities. A pie chart was produced to show the genomic distribution of AR binding sites of those 452 genes, using the genomic features obtained from the UCSC reference gene table.

The Agilent custom oligoarray was then used to determine the relative contribution of 18 clinically relevant AR-associated co-regulators to androgen-regulation of AR target gene expression. RNA from appropriate coregulator siRNA transfections was processed as above.

For each co-regulator specific siRNA transfection, we performed three separate comparisons for each gene based on the samples' characteristics: (1) level of androgen regulation by comparing control siRNA with androgen treatment versus control siRNA without treatment condition, (2) level of androgen regulation by comparing specific siRNA-silencing with androgen treatment versus specific siRNA-silencing without androgen treatment condition, and (3) fold androgen regulation in (1) compared to (2). The limma program was used to assess differential expression in each of the above-mentioned comparison. The genes with FDR <0.05 and at least 2.0 fold change are considered as differentially expressed genes.

To visualize the androgen regulation signatures upon loss of each co-regulator, we generated a heatmap by using the R package gplots. R package WGCNA was further used to generate the heatmap to display the Pearson correlation among co-regulators. Functional GSEA was conducted using the java GSEA application, version 2.0. The detailed GSEA parameters were as follows: the number of permutations was 1000, and gene set size filters with minimum of 15.

The Aligent oligoarray data have been deposited in Gene Expression Omnibus under accession number GSE66722 in MIAME-compliant format.

## Illumina Beadchip data analysis

RNA was obtained and subjected to quality control as above. HumanHT-12 v4 Beadchip analysis was done by Roswell Park Cancer Institute Genomics Shared Resource as per the manufacturer's recommendations. The raw intensity of Illumina HumanHT-12 v4 gene expression array was scanned and extracted using BeadScan, with the data corrected by background subtraction in GenomeStudio module. The lumi module in the R-based Bioconducter Package was used to transform the expression intensity into log2 scale (*Du et al., 2008*). The log2 transformed intensity data were normalized using Quantile normalization function. We used the Limma program in the R-based Bioconductor package to calculate the level of gene differential expression for each comparison. Briefly, a linear model was fit to the data (with cell means corresponding to the different condition and a random effect for array), and selected contrast for each comparison was performed. For each comparison, we obtained the list of differentially expressed genes ($\geq$2 fold change) constrained by FDR <0.05.

Illumina BeadChIP data have been deposited in GEO under accession numbers GSE66977 and GSE81780 in MIAME-compliant format.

## Ingenuity pathway analysis

Datasets containing Gene Symbol identifiers and corresponding expression values were uploaded into the application. Each gene symbol identifier was mapped to its corresponding human/mouse/rat orthologue cluster in the Ingenuity Knowledge Base (*Krämer et al., 2014*). An absolute fold change cutoff of 1.4 was set to identify molecules whose expression was significantly differentially regulated (differentially expressed genes, DEGs).

The IPA Downstream Effects Analysis (DEA) was used to identify the biological functions and/or diseases that were most significant to the dataset. A right-tailed Fisher's Exact test was used to calculate a p-value determining the probability that each biological function and/or disease assigned to these data sets is due to chance alone. Furthermore, DEA was used to predict increases or decreases of these biological functions and/or diseases occurring after androgen activation by integrating the direction change of the DEGs into a z-score algorithm calculation. Functions and/or diseases with z-scores $\leq -2$ or $\geq 2$ were considered significant.

Canonical pathways analysis identified the canonical pathways from the IPA library that were most significant to the data set. The significance of the association between the data set and the canonical pathway was measured in 2 ways: (1) the ratio of the number of molecules from the data set that map to the pathway divided by the total number of molecules that map to the canonical pathway, and (2) calculation of a p value using Fisher's Exact test to determine the probability that the association between the genes in the dataset and the canonical pathway is explained by chance alone.

Upstream Analysis was used to identify the cascade of upstream transcriptional regulators (transcription factors, enzyme, cytokine, growth factor, miRNA, compound or drug) that could explain the observed gene expression changes in these datasets, by measuring an overlap in p-value with Fisher's Exact test and by measuring the activation z-score as well to infer the activation states of the predicted transcriptional regulators.

## Cistrome analyses of ARBSs TF binding site composition

Cistrome project tools were used to analyze the ARBS genomic regions of different coregulator-dependent AR target gene expression signatures for the presence of consensus TF binding sites. A heatmap was generated using the R gplots package to visualize the association of the gene signatures with the presence of TF binding sites in ARBS genomic regions from Cistrome.

## Sample preparation for mass spectrometric analysis

After 16 hr of treatment with either R1881 (5 nM) or vehicle, nuclear extracts were prepared from LNCaP cells using the nuclear extract kit (Active Motif) and samples were immunoprecipitated as described above. The eluted samples were run on 10% Bis-Tris Novex NuPAGE gels (Life Technologies) and were silver-stained according to the manufacturer's protocol (Silver stain kit, Amersham). The bands were cut into smaller pieces to minimize excess polyacrylamide and were first reduced with 100 mM DTT and further alkylated with 150 mM iodoacetamide. All bands were digested in-gel by adding 50 ng trypsin in 50 mM ammonium bicarbonate to each gel band for 16 hr at 37°C. Peptides were extracted in 50% acetonitrile with 5% formic acid and dried using speed vac. The dried pellet was resuspended in buffer A (1% acetic acid) for LC-MS analysis.

## Mass spectrometric analysis of peptide mixtures

The mass spectrometric analysis was performed using a Dionex Ultimate 3000 LC system coupled to Finnigan LTQ-Obitrap Elite hybrid mass spectrometer (Thermo Fisher Scientific) equipped with nano-electrospray ion source. Five µL volumes of the peptide mixture was resolved on a 15 cm Dionex HPLC column (75 µm) filled with 2 µm C18-resin. The peptides were loaded with buffer A and eluted with a 2% to 70% acetonitrile gradient of Buffer B (acetonitrile/0.1% formic acid) at a flow rate of 300 nl/min for 110 min. The digest was analyzed using the data dependent multitask capability of the instrument acquiring full scan CID mass spectra to determine peptide molecular weights and product ion spectra to determine amino acid sequence in successive instrument scans. The data was processed and analyzed by searching the human reference sequence database (ftp://ftp.ncbi.nlm.nih.gov/refseq/H_sapiens/) with the programs Mascot and Sequest, The search results were further analyzed using the search program X! Tandem which is bundled into the program Scaffold. The Scaffold data was filtered based on a 1.0% FDR at the protein level and 2 positively identified peptides with a peptide threshold at 0.1% FDR. The relative abundance of protein in the IP and control samples was determined by comparing the total spectral counts identified for each protein.

## Chromatin immunoprecipitation assays

ChIP and ChIP-re-ChIP analyses were done using EZ ChIP kit (EMD Millipore, Billerica, MA) per the manufacturer's instructions with minor modifications. After crosslinking, cell pellets were lysed and shearing using a Diagenode Bioruptor Plus Sonicator using 3 × 10 cycles of 1 min on and 1 min off at the medium setting at 4C. Protein G agarose was replaced by Protein G DynaBeads (Thermo-Scientific) (60 µl/reaction, no pre-clearing). For p53 ChIP, 3 p53 antibodies (DO1 (Santa Cruz), FL393 (Santa Cruz), and pAB421 (Calbiochem)) were used in combination at equal ratios and at a total amount of 2 µg antibody per reaction. Real-time RT-PCR on ChIP'ed DNA and data analysis were done as described (*Schmidt et al., 2012*). Primer sequences used to amplify ARE-containing regions

are included below. ChIP-re-ChIP experiments started from 3 times the amount of sheared DNA that was used for a regular ChIP experiment and 6 μg antibody (targeting p53 or AR) per reaction. Elution was done via incubation for 30 min at 37C in TE buffer that was supplemented with 10 mM DTT (*Wang et al., 2007*). For Re-ChIP, eluates were diluted 1/50 in dilution buffer and IP was done using 6 μg IgG and p53-targeting antibodies, or IgG and AR-targeting antibody following the EZ ChIP protocol.

## Trypan blue exclusion experiments

LNCaP cells were cultured at a density of $3 \times 10^5$ per well in 6 well plates in medium supplemented with charcoal-stripped FBS. Two days later, cells were treated with 5 nM R1881 or ethanol vehicle for 48 hr. After 48 hr, cells were harvested in PBS and equal ratio cells to 0.4% trypan blue dye was mixed and cells were counted using a Countess II FL cell counter (Life Technologies).

## Ki67 immunocytochemistry

LNCaP cells were seeded on coverslips at a density of $1.5 \times 10^5$ in medium supplemented with charcoal-stripped FBS and treated in the same way as for the trypan blue dye exclusion studies. For immunofluorescence, cells were washed twice with PBS and fixed with cold methanol for 4 min at −20C. Fixed cells were incubated in blocking solution (1% BSA in PBS) for 1 hr. The cells were then incubated for 1 hr at room temperature with 1:500 dilution of Ki67 antibody (Abcam) in 1% BSA in PBS in a humidified chamber. After washing with PBS, the cells were incubated with the fluorescently labeled Alexa fluor 488 (Cell signaling) secondary antibody for 45 min at room temperature in a humidified chamber. After incubation, cover slips were washed 5 times with PBS, dipped once in distilled water, DAPI stained and mounted in Vectashield medium (Vector laboratories). Images were acquired using an inverted EVOS FL imager (Life Technologies) at 10x magnification. For presentation purposes, images were merged and contrast enhanced using Image J 1.43.

## Immunohistochemistry

Unstained sections of tissue microarrays (TMAs) PR954, PR753, PR483 were obtained from US Biomax. Sections were baked overnight at 58°C in a dehydration oven. Deparaffinization, sections were hydrated in three separate solutions of Xylene (20 min in the first and 20 s each in the last two solutions), followed by three separate solutions of 100% ethanol (4 min in the first and 20 s in each of the last two solutions), followed by 95% ethanol for 1 min and then, rinsed in distilled water for 5 min. Sections were incubated in 3% hydrogen peroxide for 5 min to prevent endogenous peroxidase activity and washed in distilled water for 15 min. Vectastain, Universal Elite ABC kit was used for immunostaining (Cat# PK-6200). Epitopes were retrieved by boiling slides for 30 min in a citrate-based buffer provided as antigen unmasking solution (Vector Laboratories). Non-specific binding was blocked with normal horse serum for 30 min. Affinity-purified, polyclonal rabbit antibody against PGAM (Abcam, cat # ab126534, 1:500) was used as the primary antibody. The incubation time for primary antibody was overnight at 37°C for 1 hr. A diaminobenzidine (DAB) substrate kit (Vector Laboratories; cat # SK-4100) was used for visualization according to manufacturer's instruction. The slides were counterstained with hematoxylin QS solution (Cat# H-3404, Vector Laboratories) and rehydrated by immersion into 80%, 95% and 100% ethanol followed by dipping in Xylene twice each time for two minutes.

The intensity and distribution of positive staining was evaluated. A standard 4-point scale was employed for intensity, with cores being scored as negative (no staining), 1 + (weak staining), 2+ (moderate staining) and 3+ (strong staining). The distribution of positive staining was evaluated as percentage of positive cells (0–100). The vast majority of cores stained homogeneously (100%).

## Statistical considerations

For pair-wise point comparison of categorical variables, Fisher's exact test was used. For pair-wise comparisons involving continuous variables, Student's t-test was used. For populations with unequal variances, Welch's t test was used. If normality is not satisfied (Kolmogorov-Smirnov test) even after log-transformation, wilcoxon rank sum test was used. For multi-group comparisons, ANOVA analysis was performed. To derive the statistical significance of the overlaps between lists of genes, hypergeometric tests were used. All tests were two-tailed, with p=0.05 significance cutoff.

Each microarray experiment was performed in triplicate (i.e., three times with independently isolated samples). To confirm the data obtained by microarray analysis, we verified a subset of the target gene changes by independent real time RT-PCR analysis. To eliminate potential batch effects, the samples were randomly assigned to different plates using the OSAT program to ensure that the distribution of sample groups was even across plates.

## Antibodies used

Antibodies used for these studies include AR (N-20, SantaCruz Biotechnologies, for ChIP), AR (441, SantaCruz Biotechnologies, for immunoblotting), AR (PG21, EMD Millipore, for ChIP-re-ChIP), AIB1 (NCOA3, 39797, Active Motif), BRG1 (SMARCA4, G-7, SantaCruz Biotechnologies), β-actin (4967L, Cell Signaling), IRF1 (8478S, Cell Signaling), MEP50 (WDR77, 2823S, Cell Signaling), p300 (C-20, SantaCruz Biotechnologies), p53 (DO-1, SantaCruz Biotechnologies), p53 (FL-393, SantaCruz Biotechnologies), p53 (pAB 421, Calbiochem), PGAM5 (ab126534, abcam), pHistone H2a.z (S139, Cell Signaling), PKN1 (610686, BD Biosciences), STAT3 (12640S, Cell Signaling), TIF2 (NCOA2, 610984, BD Biosciences) and Ki67 (abcam, ab15580).

## Primers used

Multiple primers were designed and used for these studies. For real-time RT-PCR, primers include AGR2 (F: TGTTTGTTGACCCATCTCTGACA and R: TCTTCAGCAACTTGAGAGCTTTC), AOF2 (F: CCACAACAGACCCAGAAGGT and R: CTGGGTGGACAAGCACAGTA), ARHGAP11A (F: GGTTCCCTTGGATGATCTGA and R: TGGTCTCCTAAGGACCCTGTT), ATP11A (F: AGGGAGAACCACATCGAAAG and R: CGAAGAATCTGCTCCTTTGC), BAG1 (F: GCAGCAGTGAACCAGTTGTC and R: CAACGGTGTTTCCATTTCCT), CAMKK2 (F: GTCTCACCACGTCTCCATCA and R: GCCAACTTGACGACACCATA), CAV1 (F: CCACCTTCACTGTGACGAAA and R: CCCAGATGTGCAGGAAAGAG), CTNNB1 (F: GCTTGGTTCACCAGTGGATT and R: GTTGAGCAAGGCAACCATTT), FHL2 (F: GGTACCCGCAAGATGGAGTA and R: CTCATAGCAGGGCACACAGA), GAK (F: CAGCAGAAGGTGTGGAGTCA and R: CTCGGGGACAGGTTGTAGAC), GNB4 (F: GGGAAGGGTAAACGTGTTAGATT and R: GCCACTGTACAAATAGAGGAATGA), GNL1 (F: CACCCCACAGGACCCTAGTA and R: GCTGCTCAAGTCCACTTTCC), GUCY1A3 (F: GATTCTTCCCGGCATCATAA and R: GATTCACAAACTCGCTGCAA), HES6 (F: CTGCCGGCTACATCCAGT and R: ATGGACTCGAGCAGATGGTT), HIP1 (F: CCAGCGGAAGACTCAAGAAC and R: CTGACTGGGCAGAAGTTTCC), HTATIP2 (F: GAAACAGAAGCCCTGTCGAA and R: CAATGAGCGTGACTTTGGAA), IRF1 (F: GGATTCCAGCCCTGATACCT and R: CACCTCCAAGTCCTGCATGT), KAT5 (F: CAGATCACACTCCGCTTCAA and R: CACTGGAGTTGCTGGTGAAA), MEF2A (F: AGCACATTGTGGGAGAGAGACTGA and R: TGGCTTGGCCATTTTTCCTGAGCA), MPRIP (F: GGTTTGCAGCAATGGAAGAA and R: CTTCGATGGCTGAGATGGTG), NCOA1 (F: CTCTGGATTCAGGGCTTCTG and R: GTTCGGCAGTTGTTGTCAAA), NCOA2 (F: GGCAAGAAGAGTTCCCATGA and R: CTGCTCTCATGGTGCTGGTA), NCOA3 (F: CACATGGGAGTCCTGGTCTT and R: GGTTCCCAGTATTGCCAGAA), NET1 (F: CTGTGGTCAGAGATGCTGGA and R: GGGTCATGGTAGGCCTTTCT), PARK7 (F: TGGCTAAAGGAGCAGAGGAA and R: ATGACCACATCACGGCTACA), PGAM5 (F: GCAAAGTCAGCACAGATCTG and R: CATCTGCGCGGTGGATGTAG), PKN1 (F: GCCATCAAGGCTCTGAAGAA and R: GTCTGGAAACAGCCGAAGAG), RAB27A (F: CTGCCAATGGGACAAACATA and R: CCGTAGAGGCATGACCATTT), RAD9A (F: GTGCGGAAGACTCACAACCT and R: CAGGAGAGAAGGGCAGAACA), RALB (F: CTACGCAGCCATTCGAGATA and R: CGGAGAATCTGTTCCCTGAA), RCHY1 (F: CCGTGTTGTTGCTCATGTCT and R: CATCATCCAGCTGTCTCCAA), SASH1 (F: TCCGAAAGAACCAGAAAGGA and R: TAGCTGAATCCGCTCCTCAT), SDC4 (F: CCACCGAACCCAAGAAACTA and R: GCACAGTGCTGGACATTGAC), SERPINB5 (F: CCCTATGCAAAGGAATTGGA and R: CAAAGTGGCCATCTGTGAGA), SMAD3 (F: CTCCAAACCTATCCCCGAAT and R: CGCTGGTTCAGCTCGTAGTA), SMARCA4 (F: CCTGAATGAGGAGGAAACCA and R: GCAGACATGTCGCACTTGAT), SMARCC1 (F: GCGGATGCTCCTACCAATAA and R: CACTTTGCAGGGAGTTTGGT), STAT3 (F: GGCCATCTTGAGCACTAAGC and R: CGGACTGGATCTGGGTCTTA), TP53 (F: GAAGACCCAGGTCCAGATGA and R: CTGGGAAGGGACAGAAGATG), WASF3 (F: CAGCTGAGCAGTCTGAGCAA and R: CTGGGTGACTTTGACAGCAA), WDR77 (F: GTCTTGAGCTCTGGCACACA and R: CAGCATGAGCTCGGTATGAA), and ZIC2 (F: GGCACCTTGTGATCATGTTG and R; CAAAGACTCCGGAAGGGATA).

For ChIP assays, primers were CAMKK2 (F: AGAACACTGTAGCTCACACAGGCA and R: GGGCACTTCCCAACCTTTCTTACT), GNB4 (F: TATGAGTCCGTCTCAGTGTTG and R: TTTGAA TGCACCTAATCAGCC), MEF2A (F: TTGTTCTGTTTCTAGTGCTGTG and R: GCCAAATCTTTCCAAG TAGC), NET1 (F: CCGAAAGTCAGCTCAGATCA and R: TTGCCTGTTCCTTCTCTCTGA), RAB27A (F: TCCTGACCACAATCATAGGTTA and R: CGTTAAAAGCAAAGTCAAGGTC), RALB (F: TAGGTGG TGGTGCTTGAGTG and R: TCTTCAGTCACAATCCTTGGAA) SASH1 (F: CATTTCAGAACAACAGGC TCAG and R: TTGTTCATTGAGGTCAACGTG) and PSA (F: ACAGACCTACTCTGGAGGAAC and R: AAGACAGCAACACCTTTTT).

For site-directed mutagenesis of p53, primers used were: T125M (F: GCCAAGTCTGTGACTTGC **ATG**TACTCCCCTGCCCTCAACAAG and R: CTTGTTGAGGGCAGGGGAGTA**CAT**GCAAGTCACA-GACTTGGC), F134L (F: CCTGCCCTCAACAAGATG**TTA**TGCCAACTGGCCAAGAC and R: GTC TTGGCCAGTTGGCA**TAA**CATCTTGTTGAGGGCAGG), C135Y (F: GCCCTCAACAAGATGTTTTAC-CAACTGGCCAAGACCTGCC and R: GGCAGGTCTTGGCCAGTTGGTAAAACATCTTGTTGAGGGC ), P152L (F: TGGGTTGATTCCACACCC**CTG**CCCGGCACCCGCGTCCG and R: CGGACGCGGG TGCCGGG**CAG**GGGTGTGGAATCAACCCA), R174W (F: ACATGACGGAGGTTGTG**TGG**CGC TGCCCCCACCATGAG and R: CTCATGGTGGGGGCAGCG**CCA**CACAACCTCCGTCATGT), V272L (F: GGACGGAACAGCTTTGAG**TTG**CATGTTTGTGCCTGTCCTG and R: CAGGACAGGCACAAACA TG**CAA**CTCAAAGCTGTTCCGTCC), R273C (F: GAACAGCTTTGAGGTG**TGT**GTTTGTGCCTGTCC TGGG and R: CCCAGGACAGGCACAAAC**ACA**CACCTCAAAGCTGTTC), N239T (F: CTACAAC TACATGTGTACCAGTTCCTGCATGGGCGGCATG and R: CATGCCGCCCATGCAGGAACTGG TACACATGTAGTTGTAG), and P359S (F: TGCCCAGGCTGGGAAGGAG**TCA**GGGGGGAGCAGGGC TCAC and R: GTGAGCCCTGCTCCCCCC**TGA**CTCCTTCCCAGCCTGGGCA).

## Replication

All experiments were performed at least twice.

## Acknowledgements

The authors thank Dr. Cassandra Talerico for helpful discussions and review of the manuscript, and Dr. Natalya Guseva for providing the LNCaP cell line in which expression of p53 has been silenced.

## Additional information

### Funding

| Funder | Grant reference number | Author |
| --- | --- | --- |
| Prostate Cancer Foundation | | Hannelore Heemers |
| National Cancer Institute | CA166440 | Hannelore Heemers |
| Velosano3 | | Hannelore Heemers |
| National Cancer Institute | 1S10RR031537-01 | Belinda Willard |

The funders had no role in study design, data collection and interpretation, or the decision to submit the work for publication.

### Author contributions

Song Liu, Formal analysis, Supervision, Writing—original draft, Writing—review and editing, Final approval of manuscript; Sangeeta Kumari, Data curation, Formal analysis, Investigation, Writing— review and editing, Final approval of manuscript; Qiang Hu, Varadha Balaji Venkadakrishnan, Dan Wang, Xiwei Chen, Changshi Lao, Jianmin Wang, Formal analysis, Investigation, Writing—review and editing, Final approval of manuscript; Dhirodatta Senapati, Adam D DePriest, Simon E Schlanger, Salma Ben-Salem, Malyn May Valenzuela, Shaila Mudambi, Wendy M Swetzig, Mojgan Shourideh, Neelu Yadav, Data curation, Investigation, Writing—review and editing, Final approval of manu-script; Belinda Willard, Conceptualization, Data curation, Investigation, Writing—review and editing, Final approval of manuscript; Gokul M Das, Data curation, Formal analysis, Supervision, Writing— review and editing, Formal approval of manuscript; Shahriah Koochekpour, Cristina Magi-Galluzzi,

Data curation, Formal analysis, Supervision, Writing—review and editing, Final approval of manuscript; Sara Moscovita Falzarano, Jean-Noel Billaud, Data curation, Formal analysis, Writing—review and editing, Final approval of manuscript; Hannelore V Heemers, Conceptualization, Data curation, Supervision, Funding acquisition, Visualization, Methodology, Writing—original draft, Project administration, Writing—review and editing, Final approval of manuscript

### Author ORCIDs
Hannelore V Heemers (ID) http://orcid.org/0000-0001-9137-5083

### Decision letter and Author response
Decision letter https://doi.org/10.7554/eLife.28482.034
Author response https://doi.org/10.7554/eLife.28482.035

## Additional files

### Supplementary files
• Supplementary file 1. Design of oligoarray, overview of AR target genes studied, and overview of coregulators considered for analysis. (A) Overview of genes included in custom Agilent oligoarray Rows, categories of genes included on 8 × 15K custom Agilent oligoarray. Columns, Number of genes identified for inclusion on the array, and number of genes for which Agilent catalogue probes were available for inclusion. (B) Overview of 452 AR target gene signature Gene name, HUGO gene symbol; FC, fold change (C) Overview of coregulators considered, prioritized and withheld for analysis A PudMed search for papers that contain the terms 'AR' and 'CaP' in their title and/or abstract was performed. Abstracts fulfilling these criteria were screened for reference to coregulator function, and if so, full-length papers were reviewed individually to verify description of a *bona fide* AR-associated coregulator. Left to right: Column 1: 181 coregulators for which literature search was done. Column 2: 51 coregulators for which differential protein expression has been reported in CaP when compared to benign prostate (yes entries). Column 3: 22 coregulators for which differential expression in CaP correlated with aggressive disease, and were analyzed in *Figures 4–6* (yes entries). Column 4: 18 coregulators for which siRNA-mediated silencing did not affect AR expression, CaP cell morphology or CaP cell survival and were included in final analyses (yes entries).
DOI: https://doi.org/10.7554/eLife.28482.019

• Supplementary file 2. Characterization of 452 AR target gene signature (A) Androgen regulation of AR target gene expression in VCaP cells VCaP cells were seeded in medium supplemented with charcoal-stripped FBS (CSS). 2 days later, medium was changed and cells were treated with 5 nm R1881 or ethanol vehicle for 48 hr. Cells were harvested and AR target gene expression was evaluated using real-time RT-PCR. Target gene mRNA levels were normalized with the values obtained from GAPDH expression and are expressed as relative expression values, taking the value obtained from one of the vehicle-treated samples as 1. *Columns*, means of values obtained from three independent biological replicates; *bars*, sem. Note the consistency of androgen regulation of majority (7/8) of AR target genes obtained from LNCaP cells also in VCaP cells. (B) Kinetics of androgen regulation of AR target gene expression LNCaP cells were seeded in medium supplemented with charcoal-stripped FBS (CSS). 2 days later, medium was changed and cells were with 5 nm R1881 or ethanol for 4 or 8 hr. Timepoints were chosen as androgen regulation of androgen-induced AR target genes is detectable at 4 hr, and androgen regulation of androgen-repressed AR target genes becomes apparent at 8 hr. Target gene mRNA levels were normalized with the values obtained from GAPDH expression and are expressed as relative expression values, taking the value obtained from one of the vehicle-treated samples at the 4 hr time point as 1. *Columns*, means of values obtained from three independent biological replicates; *bars*, sem values. Note the consistency in kinetics of androgen regulation of newly recognized AR target genes (bottom 2 rows) with that of the well-known ARE-driven genes PSA, SCAP, FN1 and SERPINB5 (top row). (C) Representative real-time RT-PCR validation of Agilent oligoarray data Side-by-side comparison of Agilent oligoarray data and real-time RT-PCR data using same RNA samples. Expression data shown are derived from genes for which Agilent oligoarray expression was low (Y axis values, top row), and thus likely less reliable.
DOI: https://doi.org/10.7554/eLife.28482.020

• Supplementary file 3. Ingenuity pathway analyses on AR target gene signatures
DOI: https://doi.org/10.7554/eLife.28482.021

• Supplementary file 4. Impact of silencing of 22 clinically relevant coregulators on prostate cancer cells and AR expression (A) Real-time RT-PCR validation of siRNA-mediated coregulator knock-out efficiency LNCaP cells were transfected with On Target siRNA SmartPools directed against 22 coregulators, against AR, or with non-targeting On Target Plus control SmartPool. At 96 hr after transfection, cells were harvested and RNA was extracted for real-time RT-PCR analysis. Target gene mRNA levels were normalized to GAPDH expression and are expressed as relative expression, taking the value obtained from one of the control siRNA-transfected conditions as 1. *Columns*, means of values obtained from three independent biological replicates; *bars*, sem. Gray bars, control siRNA-transfected cells; Black bars, specific siRNA-transfected cells (B) Morphology of LNCaP cells following coregulator silencing LNCaP cells were transfected with individual On Target siRNA SmartPools directed against each of the 22 coregulators, against AR, or a non-targeting On Target Plus control SmartPool. At 96 hr after transfection, cells were imaged using Infinity Analyze software. (C) Effects of coregulator silencing on AR protein expression levels LNCaP cells were transfected with individual On Target siRNA SmartPools directed against each of the 22 coregulators or against AR, or with non-targeting On Target Plus control SmartPools (c). At 96 hr after transfection, cells were harvested and total protein extracts were subjected to western blot analysis using an antibody directed against AR. To control for potential loading differences, blots were stripped and reprobed with an antibody targeted at β-actin.
DOI: https://doi.org/10.7554/eLife.28482.022

• Supplementary file 5. Impact of coregulator silencing on androgen regulation of AR target genes (A) Nomenclature used to describe the effect of loss of coregulator on androgen regulation of AR target gene expression co+, androgen regulation is more pronounced (+) after loss of coregulator expression and the direction of androgen regulation remains consistent (co); co-, androgen regulation is less pronounced (-) after loss of coregulator expression and the direction of androgen regulation remains consistent (co); op+, androgen regulation is more pronounced (+) after loss of coregulator expression and the direction of androgen regulation is opposite (op); op-, as androgen regulation is less pronounced (-) after loss of coregulator expression and direction of androgen regulation is opposite (op). Androgen-induced, gene which expression is increased in control siRNA (c)-transfected cells after androgen stimulation; androgen-repressed gene which expression is decreased in control siRNA (c)-transfected cells after androgen stimulation. Spec, specific for one coregulator (B) Overview of coregulators that affect androgen regulation of the genes encoding RAB27A and GNB4 (C) Experimental validation of recruitment of 6 representative coregulators to the genes encoding RAB27A and GNB4 ChIP studies verifying recruitment of 6 coregulators at AREs within the genes encoding RAB27A and GNB4. ChIP was done on LNCaP cells that had been treated with 5nM R1881 or vehicle for 16 hr. Coregulators analyzed via ChIP were chosen based on availability of suitable antibodies. (D) Time course of androgen-dependent recruitment of AR (left), NCOA3 (middle) and WDR77 (right) to AREs in the genes encoding RAB27A and GNB4 ChIP was done on LNCaP cells that had been treated with 5nM R1881 or vehicle for 1, 4, 16 or 48 hr. (E) Impact of knock-down of WDR77 or NCOA3 on kinetics of androgen-regulation of RAB27A and GNB4 LNCaP cells were transfected with On Target siRNA SmartPools directed against WDR77 or NCOA3, or with non-targeting On Target Plus control SmartPool (c). At 42 hr after transfection, cells were treated with 5nM R1881 (+) or vehicle (-). After 1, 4, 16 or 48 hr, cells were harvested and RNA was extracted for real-time RT-PCR analysis. Target gene mRNA levels were normalized to GAPDH expression and are expressed as relative expression, taking the value obtained from one of the control siRNA-transfected conditions for each siRNA group at each time point as 1. *Columns*, means of values obtained from three independent biological replicates; *bars*, sem. Gray bars, control siRNA-transfected cells; Black bars, specific siRNA-transfected cells. (Top left) Effects of WDR77 (top) and NCOA3 (bottom) silencing on GNB4 mRNA expression; (Top right) Effects of WDR77 (top) and NCOA3 (bottom) silencing on RAB27A mRNA expression: (Bottom panels) verification of WDR77 siRNA-mediated silencing (left), verification of NCOA3 siRNA-mediated silencing (right).
DOI: https://doi.org/10.7554/eLife.28482.023

• Supplementary file 6. Isolation of a STAT3-dependent AR-IRF1 transcriptional code. (A) Co-immunoprecipitation assay shows interaction between IRF1 and STAT3 in LNCaP cells. (B) Overlap in

genome-wide IRF1- and STAT3-dependent androgen-responsive gene expression. Results reflect the effect of 48 hr treatment of LNCaP cells with 5nM R1881. R1881 or vehicle treatment was administered 42 hr after siRNA transfection. Numbers, number of genes. (C) Overlap in IPA functional annotation between the androgen-responsive genes that depend on WDR77 and p53 (from *Figure 3*, panel F) and the androgen-responsive genes that depend on IRF1 and STAT3 (from panel B of this figure) (p<2.2.E-16). Genes subjected to IPA are those listed in the center of Venn diagrams shown in panel F of *Figure 3* and panel B of this figure. Numbers, number of functional annotations.
DOI: https://doi.org/10.7554/eLife.28482.024

• Supplementary file 7. cBio Portal analysis of copy number variation and somatic mutations that affect the 18 identified coregulators in patients prostate cancer specimens. n = number of CaP clinical specimens for which DNA sequencing or copy number variation data is available. Numbers in columns indicate the percentage of CaP cases in each study for which somatic mutations and/or copy number variations that affect corresponding coregulator have been reported.
DOI: https://doi.org/10.7554/eLife.28482.025

• Supplementaty file 8. TP53 binding motif identified from the promoters of WDR77- and p53-dependent androgen-responsive genes.
DOI: https://doi.org/10.7554/eLife.28482.026

• Transparent reporting form
DOI: https://doi.org/10.7554/eLife.28482.027

### Major datasets

The following datasets were generated:

| Author(s) | Year | Dataset title | Dataset URL | Database, license, and accessibility information |
|---|---|---|---|---|
| Heemers HV, Wang D | 2016 | Development of coregulator-dependent androgen receptor target gene signatures | https://www.ncbi.nlm.nih.gov/geo/query/acc.cgi?acc=GSE66722 | Publicly available at the NCBI Gene Expression Omnibus (accession no: GSE66722) |
| Heemers HV, Chen X, Wang D, Liu S | 2016 | Contribution of WDR77 and p53 to androgen response in prostate cancer cells | https://www.ncbi.nlm.nih.gov/geo/query/acc.cgi?acc=GSE66977 | Publicly available at the NCBI Gene Expression Omnibus (accession no: GSE66977) |
| Heemers HV, Chen X, Wang D, Liu S | 2017 | Contribution of STAT3, IRF1 and PGAM5 to androgen response in prostate cancer cells | https://www.ncbi.nlm.nih.gov/geo/query/acc.cgi?acc=GSE81780 | Publicly available at the NCBI Gene Expression Omnibus (accession no: GSE81780) |

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
