## [Decision Letter]

Thank you for submitting your article "Coregulator-dependency differentiates between distinct mechanisms of androgen receptor action in prostate cancer" for consideration by *eLife*. Your article has been reviewed by three peer reviewers, one of whom is a member of our Board of Reviewing Editors, and the evaluation has been overseen by Kevin Struhl as the Senior Editor. The following individual involved in review of your submission has agreed to reveal his identity: Haojie Huang (Reviewer #2).

The reviewers have discussed the reviews with one another and the Reviewing Editor has drafted this decision to help you prepare a revised submission.

Summary:

General assessment: The Reviewers felt that this work was a comprehensive and ambitious tour de force assessing the role of individual co-regulators in the androgen responsiveness of genes in prostate cancer and normal prostate. In this analysis the authors discover and characterize a novel complex consisting of AR-p53 and WDR77 that is influential in the regulation of a subset of androgen-responsive genes. Overall there was enthusiasm for the novelty of the conclusions of this work:

Central conclusions:

1 A subset of AR co-regulators appears to be broadly required for most androgen responsive genes.

2) There is considerable AR target gene preference and context-dependence among co-regulators for their impact on androgen responsiveness of target genes.

3) An important new complex in AR is described, consisting of AR, p53 and WDR77. The authors find that androgen responsiveness after p53 knock down or WDR77 knockdown affects the same subset of genes in the same direction, supporting the premise that a complex of AR-WDR77-p53 exists to control the expression of these genes.

Overall, the Reviewers felt that this study describes very novel findings regarding the complexity of the mechanisms of action of the AR in prostate cancer. In particular, it was felt that this manuscript would serve as an invaluable template for all future studies in this important and clinically-relevant area. The reviewers raised a number of concerns that must be adequately addressed before the paper can be accepted. Some of the required revisions will likely require further experimentation within the framework of the presented studies and techniques.

Essential revisions:

1) The authors state, but do not actually formally prove, that a subset of androgen-response genes is controlled by the p53-WDR77-AR complex. Specifically, in Figure 5 the authors conduct ChIP of four proteins on the RAB27A and GNB4 promoters, but this does not indicate co-recruitment on the same DNA. A ChIP-re-ChIP experiment should be done for p53 and AR on these promoters, in cells with siControl versus siWDR77, in cells +/- R1881. This experiment would be essential to prove that this complex exists on a single promoter in the presence of androgen. It would also prove that p53 and AR are brought by WDR77 to said promoter, which is an important conclusion.

2) Since the p53-WDR77 complex has not previously been reported, it seems that IP-westerns in both directions (IP p53, WB WDR77, and vice versa) in a minimum of two different cell lines is warranted, in the interest of rigor and reproducibility.

3) Appropriate statistical analyses are at times lacking. Statistical analysis for the overlapping of genes in Figure 1, Figure 3, Figure 5 should be performed and the P value for the overlapping should be provided. Statistical analysis should be run for all the comparisons between experiments shown in Figure 6 and the p values should be provided.

4) It appears that much of the cell line derived data was generated by exposing cells in stripped serum containing media with and without 5nM R1881. However, numerous publications document that cells can have a bimodal dose response to androgens, with optimal growth promotion at 0.1-1nM R1881. In contrast, in some studies 5nM R1881 has been reported to inhibit growth of cells. Thus, the authors need to document that, at least in their hands, 5nM R1881 is actually growth stimulatory. If this is not the case, their conclusions are compromised.

---

## [Author Response]

Essential revisions:1) The authors state, but do not actually formally prove, that a subset of androgen-response genes is controlled by the p53-WDR77-AR complex. Specifically, in Figure 5 the authors conduct ChIP of four proteins on the RAB27A and GNB4 promoters, but this does not indicate co-recruitment on the same DNA. A ChIP-re-ChIP experiment should be done for p53 and AR on these promoters, in cells with siControl versus siWDR77, in cells +/- R1881. This experiment would be essential to prove that this complex exists on a single promoter in the presence of androgen. It would also prove that p53 and AR are brought by WDR77 to said promoter, which is an important conclusion.

Thank you for this suggestion. We have performed ChIP-re-ChIP experiments for p53 and AR on these RAB27A and GNB4 gene regions. LNCaP cells were transfected with non-targeting control siRNA or siRNA targeting WDR77. 72 hours after transfection cells were treated with 5nM or vehicle, and 16 hours later cells were harvested and processed for ChIP-re-ChIP.

ChIP using p53 antibody followed by re-ChIP with AR antibody showed androgen-dependent enrichments on GNB4 and RAB27A genes under control conditions, confirming co-recruitment of AR and p53. This androgen-regulation was decreased after silencing of WDR77, indicating involvement of WDR77 in co-occupancy of AR and p53 on GNB4 and RAB27A genes. Similar results were obtained when ChIP was done with an antibody targeting AR, followed by re-ChIP for p53.

These results have been described in the Results section, and are included as novel panel J in Figure 5. The Materials and methods section has been updated to include a description of the experimental procedure. Please note that for these Re-ChIP studies, we used AR antibody PG21 from EMD Millipore, which has been used for AR-ChIP before (e.g. Li et al., J. Biol. Chem. 2008, 2843(43): 28988-95). Our previous AR-ChIP studies made use of AR antibody N-20 (SantaCruz Biotechology), which had been discontinued by the vendor and was no longer available in the laboratory.

2) Since the p53-WDR77 complex has not previously been reported, it seems that IP-westerns in both directions (IP p53, WB WDR77, and vice versa) in a minimum of two different cell lines is warranted, in the interest of rigor and reproducibility.

We have performed these reciprocal Co-IP experiments in LNCaP cells and VCaP cells. The experiment in LNCaP cells in which IP was done for p53 and WB for WDR77 was shown in the original Figure 3, and has been retained. The experiment using antibodies in the reverse order, i.e. IP for WDR77 and WB for p53, has been included as a novel panel C in Figure 3. The experiment in which IP was done for p53, and WB for WDR77 in VCaP cells has been included as novel panel D for Figure 3; and the experiment in which IP was done for WDR77 and WB for p53 has been included as a new panel E in Figure 3.

All 4 experiments corroborate the presence of WDR77 and p53 in same immunocomplex, in both LNCaP and VCaP cells. They have been described in the Results section of the revised manuscript, and the legends of Figure 3 have been adjusted.

In light of minor comment 6, we have moved the original panels D-F of Figure 3, which relate to the AR-STAT3-IRF1 transcriptional code, to the Supplemental Information, where it has been included as novel Supplementary file 6. Since the novel panels D-E from Figure 3 confirm and now duplicate data shown originally in panel E of Figure 4, we have omitted panel E from Figure 4 from this revised manuscript.

3) Appropriate statistical analyses are at times lacking. Statistical analysis for the overlapping of genes in Figure 1, Figure 3, Figure 5 should be performed and the P value for the overlapping should be provided. Statistical analysis should be run for all the comparisons between experiments shown in Figure 6 and the p values should be provided.

We apologize for this oversight. The requested statistical analyses have been performed and Figure 1, Figure 3 and Figure 5, the text of the Results section, and the respective legends have been adjusted to include these data. The Materials and methods section has been updated to include a description of the statistical methods used for these comparisons.

For Figure 1 new Source data file “Figure 1—source data 2” has been included, which lists the p-values for each of the 136 comparisons in the same format as Figure 1.

For Figure 6, significant changes with p<0.05 have been marked in the figure using the symbol “*” in the graphs for panels A, B, C, D and E. A new source data file – Figure 6—source data 1 – has been added also which lists the p-values for all comparisons shown in Figure 6.

A description of the statistical methods used has been included in the legends to Figure 1—source data 2, Figure 6—source data 1, and in the revised Materials and methods section.

4) It appears that much of the cell line derived data was generated by exposing cells in stripped serum containing media with and without 5nM R1881. However, numerous publications document that cells can have a bimodal dose response to androgens, with optimal growth promotion at 0.1-1nM R1881. In contrast, in some studies 5nM R1881 has been reported to inhibit growth of cells. Thus, the authors need to document that, at least in their hands, 5nM R1881 is actually growth stimulatory. If this is not the case, their conclusions are compromised.

Thank you for the suggestion. We have addressed this comment by first performing trypan blue exclusion assays on LNCaP cells that had been treated for 48 hours with either 5nM R1881 or vehicle. A ~2-fold increase in cell proliferation was found after androgen treatment. These data have been included in a novel Figure 1—figure supplement 2, panel A. These finding have been validated further in Ki67 immunofluorescence staining assays on LNCaP cells that had been treated for 48 hours with 5nM R1881 or vehicle. Cell counts of positively staining cells were done, which showed an increase in the percentage of Ki67-positive nuclei in the R1881-treated condition compared to the vehicle-treated condition. These data have been included as novel Figure 1—figure supplement 2, panel B. Representative images of vehicle- or 5nM R1881-treated cells have been included also as novel Figure 1—figure supplement 2, panel C. These experiments confirm the growth stimulation of LNCaP cells under these experimental conditions, which is consistent also with the cell viability data in panel B of Figure 6.